# Mediator complex interaction partners organize the transcriptional network that defines neural stem cells

Marti Quevedo[1], Lize Meert[1], Mike R. Dekker[1], Dick H.W. Dekkers[2], Johannes H. Brandsma[1], Debbie L.C. van den Berg ⓘ [1], Zeliha Ozgür[3], Wilfred F.J. van IJcken ⓘ [3], Jeroen Demmers[2], Maarten Fornerod[1] & Raymond A. Poot[1]

The Mediator complex regulates transcription by connecting enhancers to promoters. High Mediator binding density defines super enhancers, which regulate cell-identity genes and oncogenes. Protein interactions of Mediator may explain its role in these processes but have not been identified comprehensively. Here, we purify Mediator from neural stem cells (NSCs) and identify 75 protein-protein interaction partners. We identify super enhancers in NSCs and show that Mediator-interacting chromatin modifiers colocalize with Mediator at enhancers and super enhancers. Transcription factor families with high affinity for Mediator dominate enhancers and super enhancers and can explain genome-wide Mediator localization. We identify E-box transcription factor Tcf4 as a key regulator of NSCs. Tcf4 interacts with Mediator, colocalizes with Mediator at super enhancers and regulates neurogenic transcription factor genes with super enhancers and broad H3K4me3 domains. Our data suggest that high binding-affinity for Mediator is an important organizing feature in the transcriptional network that determines NSC identity.

[1] Department of Cell Biology, Erasmus MC, Wytemaweg 80, 3015 CN Rotterdam, Netherlands. [2] Center for Proteomics, Erasmus MC, 3015 CN Rotterdam, Netherlands. [3] Center for Biomics, Erasmus MC, 3015 CN Rotterdam, Netherlands. Correspondence and requests for materials should be addressed to R.A.P. (email: r.poot@erasmusmc.nl)

The Mediator complex is a complex of ~30 subunits that is important for transcriptional regulation and is conserved from yeast to human[1–4]. The Mediator complex provides communication between active enhancers and promoters by interacting with proteins that bind to either of these two classes of regulatory DNA elements[2,3,5]. Accordingly, identified Mediator-interacting proteins include many transcription factors[2,5], RNA polymerase II (RNApol2) and transcription elongation factors[6]. Recently, Mediator content was used to rank enhancers in embryonic stem cells (ESCs) and enhancers with the highest Mediator content were postulated as super enhancers (SEs)[7], a class of enhancers that regulates key genes in cell identity and oncogenes[7–9]. Related enhancer types such as stretch enhancers and anti-pause enhancers were described independently[10,11]. There is debate on whether SEs act mechanistically different from typical enhancers[12]. Arguments in favor of the functional distinction of SEs is their ability to drive high levels of transcription and their selective sensitivity to inhibitors of Brd4, a chromatin-binding protein enriched at SEs[9,10,13]. Besides Mediator and Brd4, chromatin modifiers such as Ep300 and Kdm1a (LSD1 complex), chromatin remodelers such as Chd7, Brg1 (SWI-SNF complex) and Chd4 (NuRD complex) and Smc1a (Cohesin complex) were found to be enriched at SEs[8]. In a recently proposed model, the constituent enhancers of an SE and their regulated promoter(s) would group together to form a phase-separated assembly[14]. Such an assembly would rely on interactions between transcriptional and chromatin regulators[14].

Cell-type specific master TFs colocalize with Mediator at SEs[7,8]. However, evidence for interactions between master TFs and Mediator, which would underpin their role in recruiting Mediator to SEs, is scarce. For example, among SE-binding master TFs Oct4, Sox2 and Nanog (ESCs), Pu.1 (pro-B cells), MyoD (Myotubes) and C/EBPα (Macrophages)[7], Mediator interactions were only detected in immunoprecipitations of Sox2 and C/EBPα and these were with single Mediator subunits[15,16]. Also our understanding of the recruitment of the above chromatin modifiers to enhancers and SEs and their subsequent maintenance at high levels at SEs is far from complete. Mediator was shown to interact with SE-enriched chromatin modifier Crebbp[17] and the Cohesin complex[18], suggesting that Mediator could, in principle, provide an anchoring role at enhancers, SEs and the proposed phase-separated assemblies.

To investigate the relevance of Mediator interactors in defining enhancers and SEs, here we describe the purification of the Mediator complex from neural stem cells (NSCs) and identify its protein–protein interaction partners by mass spectrometry. To prevent recording interactions that are mediated via DNA/chromatin, we purify Mediator from non-treated nuclear extracts, nuclear extracts treated with nuclease benzonase and nuclear extracts treated with ethidium bromide to disrupt protein-DNA interactions and only take interactions with the Mediator complex that are not affected by these treatments. Our resulting Mediator interactome contains 95 proteins of which 75 have not been, to the best of our knowledge, previously characterized as Mediator-interacting proteins. Subsequently, we perform Mediator ChIP-seq in NSCs and define SEs in NSCs by their Mediator content. Remarkably, we find that the three most frequent motifs in SEs are bound by multiple members of the small set of TFs that we identify as Mediator interactors in NSCs. We show that one of these TFs, Tcf4, regulates a set of key NSC transcription factor genes with SEs and broad H3K4me3 domain-containing promoters. High-Mediator affinity therefore appears an important characteristic of master TFs. Our Mediator interactome contains many known enhancer-binding chromatin modifiers and we show that Mediator-interacting chromatin modifiers Jmjd1c and Carm1 bind genome-wide to enhancers and SEs. Together this suggests that high-Mediator-binding affinity selects proteins that play important roles in establishing and maintaining enhancers and SEs to facilitate the regulation of cell identity.

## Results

**Purification of the Mediator complex from neural stem cells.** We generated a mouse neural stem cell line expressing FLAG-tagged Med15 (F-Med15 NSCs) to enable the purification of the Mediator complex by our FLAG-affinity protocol, which combines high efficiency and low background[19] and was extensively validated in the past for accuracy by independent immunoprecipitations of endogenous proteins[19,20]. F-Med15 NSCs and parental NSCs were grown to large scale and nuclear extracts prepared (see Methods). We were interested in proteins that can bind to the Mediator complex relying solely on protein–protein interactions and not being mediated via chromatin, which may co-purify with a chromatin-binding factor such as the Mediator complex. We reasoned that proteins interacting with Mediator by protein–protein interaction would not show a reduced interaction efficiency when treating the nuclear extract with the DNA–RNA digesting enzyme Benzonase or with ethidium bromide (EtBr), which intercalates in the DNA and disrupts protein-DNA interactions, as compared to untreated nuclear extracts (Fig. 1a). The used nuclear extract preparation procedure[21] aims to minimize the amount of DNA/chromatin in the extract by gently douncing the nuclei as a method for lysis. Nevertheless, remnants of DNA/chromatin do get released from the nuclei into the extract (Fig. 1b, Untreated). Addition of benzonase completely removed chromatin/DNA from the extract. (Fig. 1b, compare Benzonase to Untreated). We purified the Mediator complex by FLAG-affinity from nuclear extracts treated with Benzonase, with EtBr or not treated, as well as from parental NSCs as a control. Purified Mediator samples and control samples were analyzed by mass spectrometry to identify the proteins present in these samples. We selected proteins that were specific for Mediator samples and that did not go down in abundance (less than two-fold drop in emPAI score) when comparing purifications from nuclear extracts treated with Benzonase or EtBr, to purifications from untreated extracts (see Methods). To be included in our final list of Mediator-interacting proteins (Fig. 1c, Supplementary Data 1), selected proteins also had to be specifically present in an independent replicate of the Mediator purification from Benzonase-treated nuclear extract (Supplementary Data 1).

**A Mediator interactome in neural stem cells.** We identified 122 Med15-interacting proteins from the four FLAG-Med15 purifications (Fig. 1c, Supplementary Data 1), of which 26 proteins are core-subunits of the Mediator complex, leaving 96 proteins that we postulate as Mediator complex-interacting proteins. The vast majority of these Mediator-interacting proteins, 77 proteins, were not previously identified as binding to Mediator (Fig. 1c, indicated in red). Mediator-interacting proteins may interact with Mediator directly or via other proteins. A number of well-known constituents of enhancers such as Ep300, Chd7, LSD1 complex, NuRD complex and SWI-SNF complex were identified as interactors of Mediator (Fig. 1c, Supplementary Data 1). Cohesin subunit Smc1a[18] was identified, whereas Cohesin subunit Smc3 and Cohesin loader Nipbl were observed in three out of four Mediator purifications and are therefore not part of the final Mediator interactor list (Supplementary Data 1). Ep300, Crebbp, Chd7, Kdm1a (LSD1 complex), Chd4 (NuRD complex), Smc1a (Cohesin) and Brg1 (SWI-SNF complex) were recently shown, like Mediator, to have higher binding densities at super enhancers (SEs) in embryonic stem cells, as compared to typical enhancers[8]. Other transcriptional activators and repressors interacting with

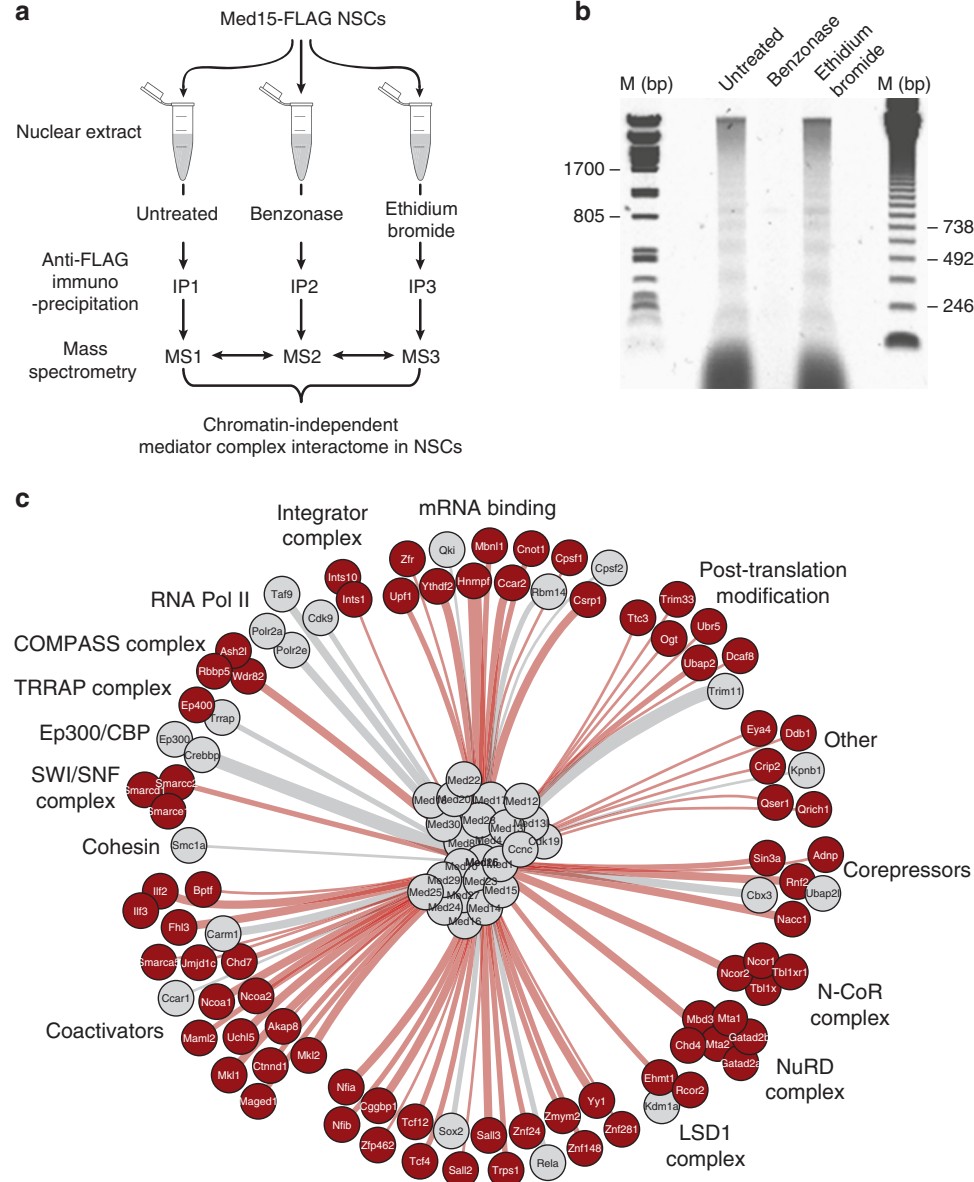

**Fig. 1** Mediator complex interactome in neural stem cells. **a** Schematic representation of Mediator complex purifications from neural stem cells (NSCs) expressing Med15-FLAG. Mass spectrometry results of the three conditions were compared to select proteins that do not decrease in abundance upon treatments as chromatin-independent Mediator complex interactors. IP immunoprecipitation, MS Mass spectrometry. **b** Agarose gel with DNA from untreated NSC nuclear extract or nuclear extract treated with Benzonase or Ethidium Bromide, as indicated. DNA size markers (M) are indicated. Source data are provided as a Source Data file. **c** Interactome of the Mediator complex in NSCs. Novel Mediator interaction partners are in red, known Mediator interaction partners are in grey. Thickness of the edges gives an indication of the relative molar protein quantity observed in the purified Mediator complex samples

Mediator included Ncoa1-2, the COMPASS complex, Integrator complex, TRRAP complex and N-CoR complex (Fig. 1c). We identified histone demethylase Jmjd1c and arginine demethylase Carm1 as Mediator interactors. Carm1 was recently identified to bind Med9 in a high throughput interaction screen[22]. We independently confirmed the interactions of Jmjd1c and Carm1 with Mediator by reverse co-immunoprecipitations with Carm1 antibodies (Fig. 2a) and Jmjd1c antibodies (Fig. 2b). One prominent Mediator interactor category is mRNA binding proteins (Fig. 1c). We find that Mediator interacts with alternative splicing regulators Hnrnpf and Mbnl1 and cleavage and polyadenylation factors Cpsf1 and Cpsf2. These interactions may facilitate the role that Mediator plays in regulating alternative splicing and alternative cleavage and polyadenylation of pre-mRNAs[23].

Mediator has been identified as a co-activator of many DNA sequence-specific transcription factors, often nuclear hormone receptors[2,24,25]. We identified 16 DNA sequence-specific transcription factors (TFs) of which 14 are novel Mediator interactors (Fig. 1c). Identified TFs include NFI TFs Nfia and Nfib, Sox2 and E-box TFs Tcf4 and Tcf12. The majority of these TFs have an important function in the regulation of NSCs (Fig. 2c). To test whether detected Mediator-interacting TFs are the highest expressed TFs in NSC, which could explain their detection by mass spectrometry, we plotted the 16 detected TFs against the 600 highest expressed TFs (by RNA-seq) in our NSCs. We find that Mediator-interacting TFs are not the highest expressed TFs in NSCs (Fig. 2d). This suggests that the detection of our Mediator-interacting TFs is primarily related to their high

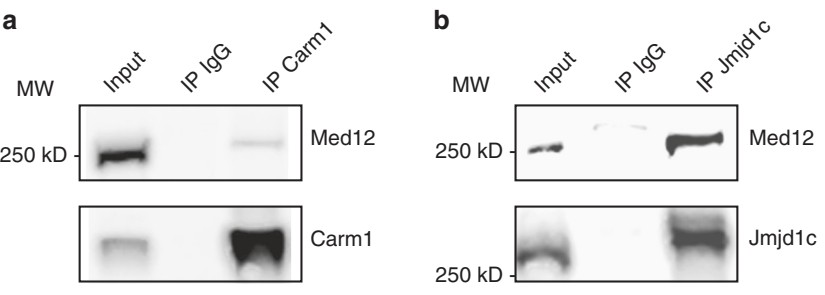

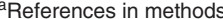

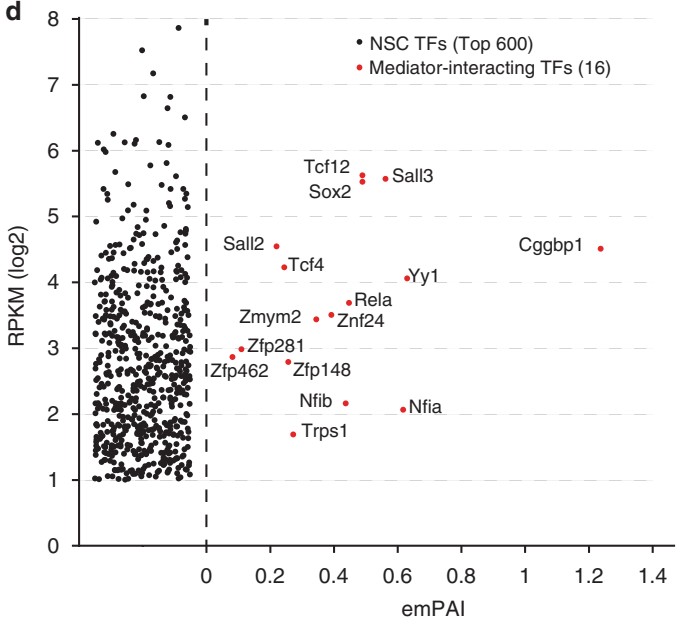

| TF name | Function in neural development[a] |
|---|---|
| Yy1 | Enhancer looping in neural stem cells |
| Nfia | Neural stem cell differentiation and glial precursor self-renewal |
| Sox2 | Neural fate specification and neural stem cell self-renewal |
| Nfib | Neural stem cell differentiation and glial precursor self-renewal |
| Sall3 | Photoreceptor and retina development and neural stem cell self-renewal |
| Tcf12 | Neural stem cell differentiation |
| Znf24 | Neural stem cell self-renewal |
| Rela | Neural stem cell self-renewal |
| Tcf4 | Neural stem cell differentiation and neural migration |

[a]References in methods

**Fig. 2** Mediator complex interactor validation. **a** Immunoprecipitation (IP) of Carm1 and Med12 by a Carm1 antibody from NSC nuclear extract. Western blots are probed with the indicated antibodies. Control IP by rabbit IgG and 5% input are also shown. Source data are provided as a Source Data file. **b** Immunoprecipitation (IP) of Jmjd1c and Med12 by Jmjd1c antibody from NSC nuclear extract. Western blots are probed with the indicated antibodies. Control IP by rabbit IgG and 5% input are also shown. Source data are provided as a Source Data file. **c** Function in neural development of identified Mediator-interacting transcription factors in NSCs. References are provided in the Methods. **d** mRNA levels in NSCs of Mediator-interacting transcription factors (TFs) and the Top 600 highest expressed TFs in NSCs. The average emPAI scores, a semi-quantitative mass spectrometry-based measure of molar amounts, in the four Mediator complex purifications is shown for Mediator–interacting TFs

binding affinity for Mediator, as compared to many other, not detected, TFs.

Brd4 has been shown to strongly colocalize with Mediator at enhancers and promoters. Despite our high sensitivity of detecting Mediator interactors, we did not detect Brd4 in any of our FLAG-Med15 purifications (Supplementary Data 1 and data not shown). We also did not detect Jmjd6 and Nsd3, functional interaction partners of Brd4[10,26], in any purification.

To validate our FLAG-affinity approach, we also purified endogenous Mediator from NSCs by immunoprecipitation with a Med12 antibody (Supplementary Data 2). We find back 60 of the 96 interactors identified in FLAG-Mediator purifications, including 11 transcription factors. With the lower sensitivity and higher background generally observed in endogenous IPs, we consider this number of overlapping Mediator interactors a validation of our FLAG-Mediator purifications.

In conclusion, we expanded the Mediator interactome with many transcription-associated factors and our experimental set-up suggests that these interactions are independent of chromatin.

**Mediator-based super enhancers in neural stem cells.** High-Mediator content is a defining feature of so-called super enhancers (SEs)[7]. SEs have not been defined yet in NSCs. We identified SEs in NSCs by ranking NSC enhancers, which were previously defined by the presence of the H3K27ac mark and Ep300[27], by their Med1 ChIP signal using the ROSE algorithm[7,9]. Accordingly, we identified 445 SEs in NSCs and assigned the 9436 remaining enhancers as typical enhancers (Fig. 3a, b, Supplementary Data 3). Transcription factors encoded by genes near top SEs include Mediator-interactors Nfia, Tcf4, Sox2 and Sall3 (Fig. 3b). We find that active genes near SEs (SE genes) in NSCs are, on average, several fold higher expressed than genes near typical enhancers (Fig. 3c). DNA motif enrichment analysis revealed that E-box, NFI and SOX motifs were the first, second and third most frequent TF DNA binding motifs in Mediator peaks, both within typical enhancers and SEs (Fig. 3d). These motifs were also previously observed in NSC enhancers defined by H3K27ac and Ep300[27]. Interestingly, TFs that can bind these motifs are well represented within the select group of TFs that we find interacting with Mediator, with Tcf4 and Tcf12 binding E-box motifs, Nfia and Nfib binding NFI sites and Sox2 binding SOX sites. In summary, we identified SEs in NSCs and find that the E-box motif is the most frequently occurring motif in Mediator peaks within typical enhancers and SEs in NSCs.

**Overlap Mediator and interaction partners outside promoters.** The identification of Mediator-binding sites in NSCs allowed us to probe its genome-wide overlap with identified Mediator interaction partners. We first focused on Mediator-interacting transcription factors, which with their sequence-specific DNA binding capacity would be candidates for Mediator-recruitment to the genome. Using published ChIP-seq datasets for TFs Nfia and Nfib (combined ChIP-seq; NFI) and Sox2[27], we found that binding sites of NFI and Sox2 highly overlap with Mediator-binding sites outside promoters, including at typical enhancers and SEs (Fig. 4a). Using our Tcf4 ChIP-seq dataset[28], we show that Tcf4 has an even higher overlap with Mediator outside promoters, at typical enhancers and at SEs (Fig. 4a), consistent with the finding that the E-box is the most frequent TF motif at Mediator-binding sites in enhancers and SEs in NSCs (Fig. 3d). The sum of binding sites of Tcf4, Sox2 and NFI (T + S + N) covers nearly 80% of all Mediator-binding sites outside promoters and over 80% of Mediator-binding sites within typical enhancers and SEs (Fig. 4a). The combined binding sites of representatives of three TF families that we find interacting with Mediator, could therefore potentially account for nearly all recruitment of Mediator outside promoters in NSCs. Examples of the overlap of Mediator with Mediator-interacting TFs are shown in Fig. 4b and c.

Subsequently, we investigated the overlap of Mediator with interacting chromatin modifiers. We performed ChIP-seq for identified Mediator-interactors arginine methylase Carm1 and H3K9 demethylase Jmjd1c. We found that Carm1 and Jmjd1c highly overlap with Mediator outside promoters, at enhancers and at SEs (Fig. 4a). Chromatin remodeler Chd7 is known to bind enhancers in ES cells[29] and indeed overlaps with Mediator at enhancers and SEs in NSCs (Fig. 4a). As expected, RNApol2 and its associated Integrator complex[30] show a high overlap with Mediator at promoters (Fig. 4a). Polycomb protein Cbx8 and insulator protein Ctcf, which we never found interacting with Mediator, show low genome overlaps with Mediator (Fig. 4a).

Examples of the overlap of Mediator with interacting chromatin modifiers are shown in Fig. 4b and c. As expected, we also find high overlaps between Mediator-interacting TFs and Mediator-interacting chromatin modifiers (Fig. 4d). We conclude that Mediator shows high binding site overlap at enhancers and SEs with interacting TFs Tcf4, NFI and Sox2 and with interacting chromatin modifiers Jmjd1c, Carm1 and Chd7.

We tested whether genome recruitment of Mediator depends on some of its interacting TFs. We performed shRNA-mediated knock-down for TFs, Tcf4 or Sox2 (Fig. 5a). We selected a number of enhancers from our ChIP-seq data for Mediator, Tcf4 and Sox2 where Mediator genome binding overlaps with genome binding by Tcf4 and Sox2. We find by Med12 ChIP RT-PCR that Mediator is indeed highly enriched at the selected sites (Fig. 5b). Knock-down of Tcf4 significantly reduced Mediator binding at all five selected sites (Fig. 5c). Knock-down of Sox2 significantly reduced Mediator binding at enhancers 6.7 kb upstream from *Olig1* and 6 kb in *Tulp3* (Fig. 5d). We find that Mediator binding at 30 kb downstream of *Olig1*, 8.6 kb in *Klf15* and 6.5 kb in *Jag1* are not significantly affected by Sox2 knock-down (Fig. 5d). We conclude that efficient Mediator recruitment to individual genomic sites can depend on its interaction partners Tcf4 or Sox2.

**Genes with SEs and broad H3K4me3 promoters in NSCs.** Recently genes with broad H3K4me3 domains at their promoters were identified[31,32], including in NSCs[31]. The top 5% of broadest H3K4me3 domains in promoters (here abbreviated as broad promoters) associated with cell-identity genes[31] and tumour-suppressor genes[32]. Mechanistically, broad promoters have increased rates of transcription elongation and higher transcriptional consistency[31,32] and show enhanced DNA looping interactions with SEs[33], compared to their typical counterparts. We found that the complete sets of SE genes and broad promoter genes in NSCs both have Transcription Regulation as their lead Gene Ontology (GO) category (Fig. 6a and Supplementary Data 4). Transcriptional regulator genes within the SE category showed neurogenesis as the only significant GO term, whereas transcriptional regulator genes within the broad promoter category included neurogenesis as one of three significant GO terms (Fig. 6a and Supplementary Data 4). The observed enrichment in transcriptional regulators acting in neurogenesis is in line with the association with cell-identity genes that has been postulated for genes with SEs[7,8] or genes with broad promoters[31]. We find that genes with broad promoters partially overlap with SE-associated genes in NSCs (Fig. 6b). Genes with SEs and broad promoters (SE + Broad) strongly enrich for transcriptional regulators acting in neurogenesis (Fig. 6b, Supplementary Data 4 and 5). Remarkably, both left-over categories of genes, genes with broad promoters but without SEs (Broad-SE) and genes with SEs but without broad promoters (SE-Broad) lose transcriptional regulators acting in neurogenesis as a GO term, whereas SE-Broad genes lose Transcriptional Regulation as a GO term altogether (Fig. 6b, Supplementary Data 4). Indeed, Mediator-interacting TFs Tcf4, Sox2, Sall3, Nfia and Nfib, as well as other well-known neural TFs including Olig1-2, Pou3f1, Pou3f3 and Npas3 and oncogene Myc have broad promoters and SEs (Supplementary Data 5). We find that SE + Broad genes are, on average, higher expressed than SE-Broad genes or Broad-SE genes, even when comparing the top 100 of each category (Fig. 6c). We conclude that in NSCs, genes with both SEs and broad H3K4me3 promoters account for the association of the separate categories of SE genes and broad promoter genes with transcriptional regulators acting in neurogenesis. Broad promoters and SEs appear to act synergistically to give higher

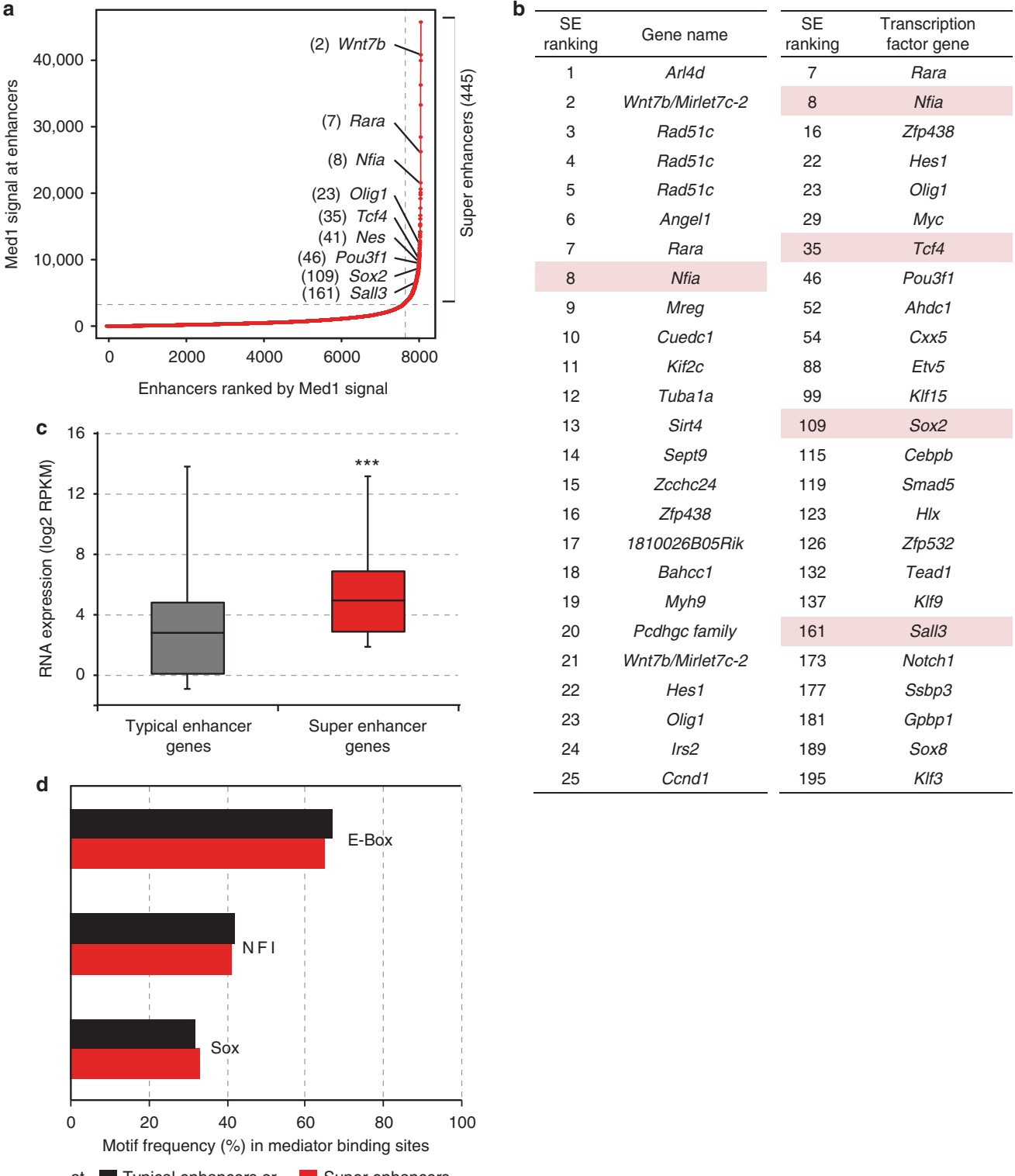

**Fig. 3** Super enhancers in neural stem cells. **a** Distribution of Med1 ChIP-seq signal (total reads) in enhancer regions in NSCs. 445 enhancers regions in the right upper quadrant are postulated as super enhancers. Examples of genes near super enhancers and the super enhancer rank are indicated. Source data are provided as a Source Data file. **b** Top 25 super enhancers (SEs) in NSCs, ranked by Mediator content, and their nearest active gene (left panel). Top 25 active transcription factor genes nearest to SEs (Right panel). SE rank is indicated. Genes encoding transcription factors that we identified as Mediator interactors are red-shaded. **c** Distribution of mRNA expression in NSCs of active genes nearest to SEs and active genes nearest to typical enhancers, but not nearest to SEs. Whiskers represent ultimate range. Significance of the difference in mRNA levels between two gene categories was assessed by Student $t$-test (***$p < 0.001$). Source data are provided as a Source Data file. **d** Most frequent transcription factor DNA motifs in Mediator-binding sites at typical enhancers and SEs. Motif frequency is indicated as the percentage of all Mediator-binding sites at typical enhancers or SEs that harbour this motif

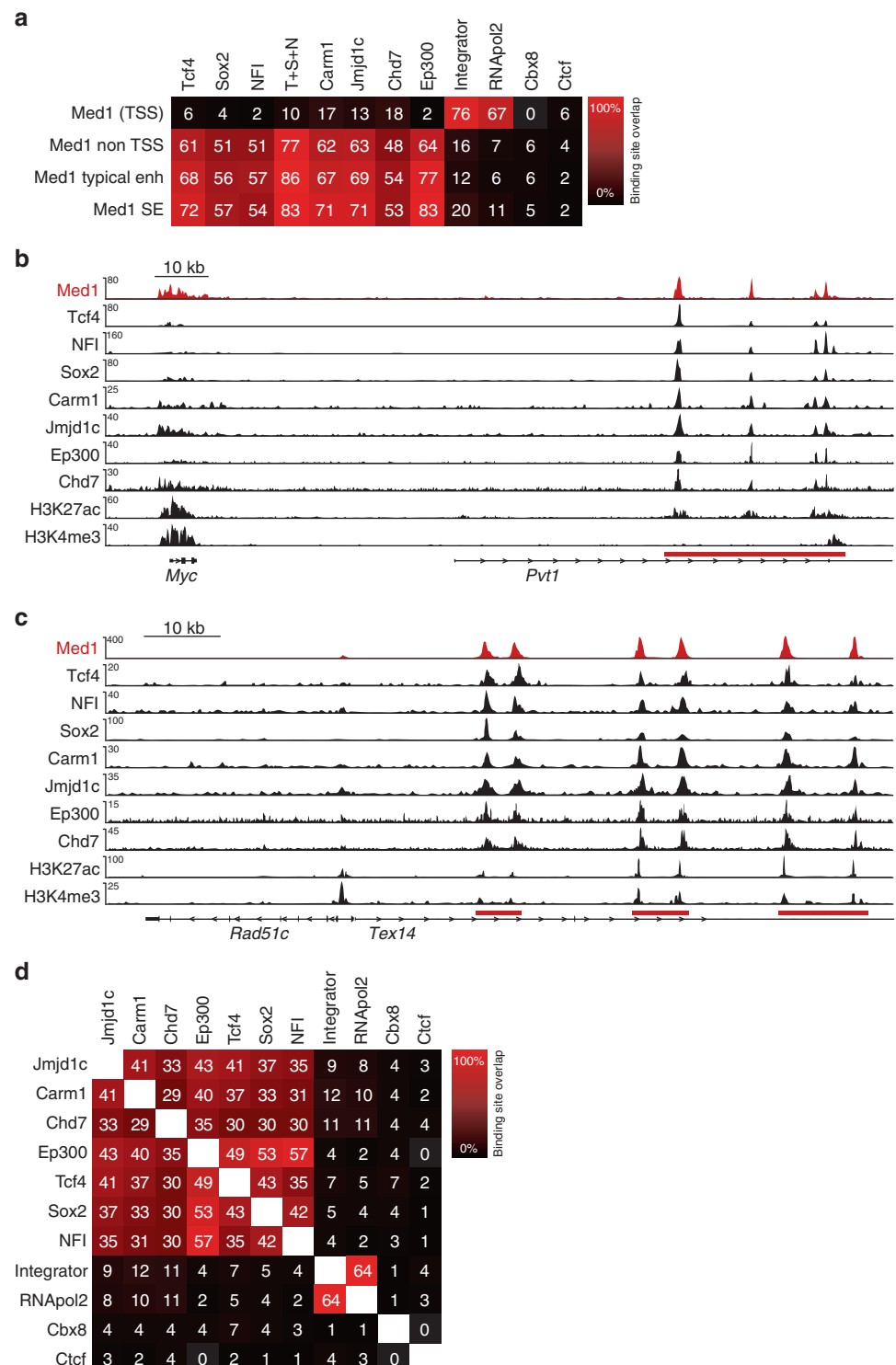

**Fig. 4** Binding site overlap of Mediator complex and its interactors. **a** Percentage overlap of genome-wide binding sites of Mediator (Med1) with Mediator-interactors Tcf4, Sox2, NFI (Nfia + Nfib), Carm1, Jmjd1c, Chd7, Ep300, Integrator complex (Ints11 subunit), and RNApol2 in NSCs. Cbx8 and Ctcf were not identified as Mediator interactors and serve as negative controls. Percentages overlap of binding sites, as determined by ChIP-seq, are indicated. T + S + N, sum of the binding sites of Tcf4, Sox2, and NFI. TSS, within 1 kb of a transcription start site. **b** Overlap of binding sites of Mediator (Med1) with binding sites of Mediator interactors at the *Myc* locus in NSCs. ChIP-seq tracks for the indicated proteins and histone modifications at the *Myc* gene are shown. The *Myc* SE in the adjacent (inactive) *Pvt* gene is indicated with a red bar. Range of reads per million per base pair is indicated on the y-axis. Scale bar is indicated. **c** Overlap of binding sites of Mediator (Med1) with binding sites of Mediator interactors at the *Rad51c* locus in NSCs. ChIP-seq tracks for the indicated proteins and histone modifications at the *Rad51c* gene are shown. The *Rad51c* SEs in the adjacent (inactive) *Tex10* gene are indicated with red bars. Range of reads per million per base pair is indicated on the y-axis. Scale bar is indicated. **d** Overlap of genome-wide binding sites of Mediator interactors and Cbx8 and Ctcf in NSCs. Percentages overlap of binding sites, as determined by ChIP-seq, are indicated

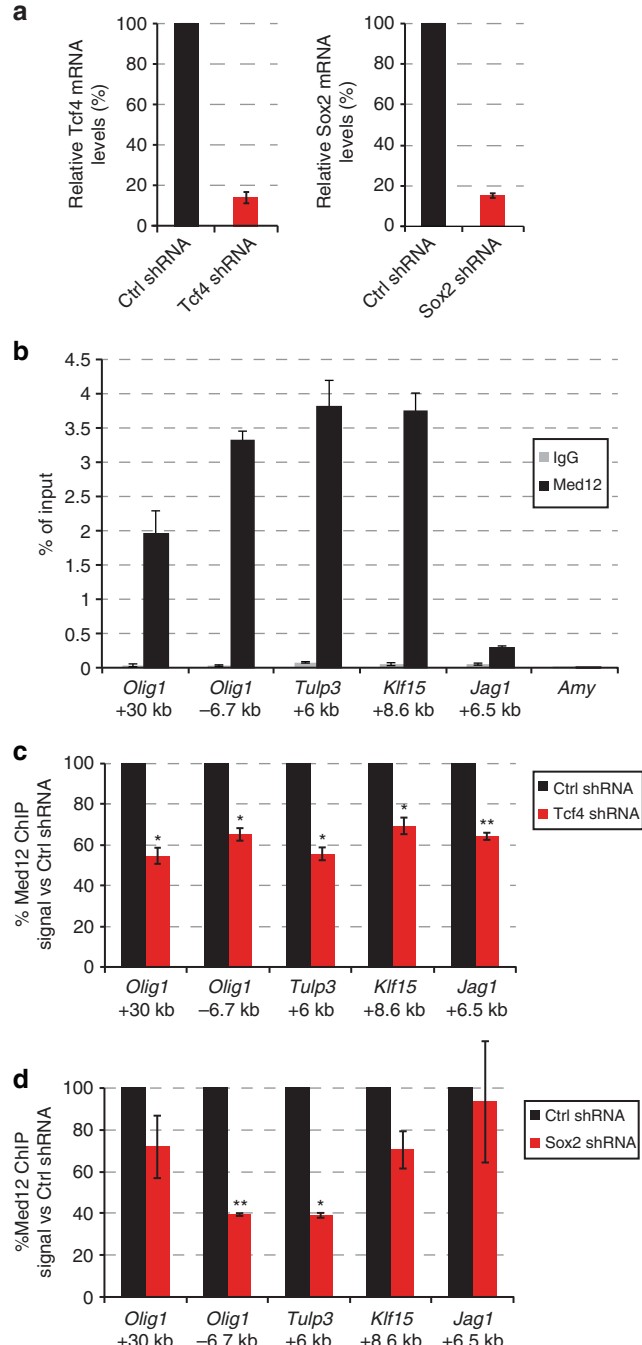

**Fig. 5** Mediator genome recruitment upon knock-down of Tcf4 or Sox2. **a** Relative mRNA levels of Tcf4 48 h after transfection with pSuper-Tcf4-shRNA or pSuper-control-shRNA (left panel), Relative mRNA levels of Sox2 46 h after transfection with pSuper-Sox2-shRNA or pSuper-control-shRNA (right panel). Source data are provided as a Source Data file. **b** Mediator ChIP signal on selected enhancers at the indicated distances from the TSS of the indicated genes. RT-PCR signals on the indicated genome areas of Med12 ChIP (Med12) and control rabbit IgG ChIP (IgG) are indicated as percentage of chromatin input. Amylase (Amy) functions as a negative control genomic region. S.e.m. is indicated of two independent experiments. Source data are provided as a Source Data file. **c** Mediator ChIP signal at selected enhancers upon knock-down of Tcf4. Med12 ChIP RT-PCR signals on the indicated genome areas in NSCs transfected with a plasmid expressing Tcf4-shRNA are indicated as percentage of the ChIP signal of NSCs transfected with a plasmid expressing control shRNA. S.e.m. is indicated of two independent experiments. Significance of the difference in Med12 ChIP signal between Tcf4-depleted NSCs and control NSCs was assessed by an unpaired Student $t$-test (*$p < 0.05$, **$p < 0.01$). Source data are provided as a Source Data file. **d** Mediator ChIP signal at selected enhancers upon knock-down of Sox2. Med12 ChIP RT-PCR signals on the indicated genome areas in NSCs transfected with a plasmid expressing Sox2-shRNA are indicated as percentage of the ChIP signal of NSCs transfected with a plasmid expressing control shRNA. S.e.m. is indicated of two independent experiments. Significance of the difference in Med12 ChIP signal between Sox2-depleted NSCs and control NSCs was assessed by an unpaired Student $t$-test (*$p < 0.05$, **$p < 0.01$). Source data are provided as a Source Data file

expression in NSCs, as compared to genes with only one of these regulatory elements.

**Binding of Mediator and interaction partners at promoters.** We investigated transcriptional regulators binding around promoters of Broad + SE genes. We found that Broad + SE genes had higher and broader promoter signals for H3K4me3, RNA-pol2 and Integrator than SE-Broad and Broad-SE genes (Fig. 6d). Mediator complex binding to promoters has not yet been analyzed genome-wide at broad promoters or genes nearest to SEs. We found that Mediator has a much higher and broader ChIP signal at Broad + SE genes than at SE-Broad, Broad-SE and typical genes (Fig. 6d). Interestingly, we observed the same for Mediator interactors T + S + N, Jmjd1c, Carm1 and Chd7 (Fig. 6d). The shape of Mediator signal tracked closely to that of

its interactors with a shoulder upstream of the TSS and a long tail into the gene (Fig. 6d). As the SE + Broad definition appears to select for genes with the broadest and highest H3K4me3 signal (Fig. 6d), we also tested the top 100 SE + Broad, top 100 SE-Broad and top 100 Broad-SE genes to have more equal signals. Indeed top 100 SE + Broad and top 100 Broad-SE have more similar H3K4me3 signals (Supplementary Fig. 1a) and showed more similar signals for Mediator and its interactors at the TSS and upstream of the TSS. However, Mediator and its interactors have a higher signal downstream of the TSS in SE + Broad genes, as compared to all other categories. Top 100 SE-Broad genes have a more narrow signal for all these factors (Supplementary Fig. 1a). The close similarity between the Mediator signal and the signals of its interactors Tcf4, Sox2, NFI, Jmjd1c, Carm1 and Chd7 is also apparent at individual broad promoter regions (Fig. 6e and Supplementary Fig. 1b). Top 100 SE + Broad promoters have more RNApol2 and Integrator signal than top 100 Broad-SE and top 100 SE-Broad promoters (Supplementary Fig. 1a), suggesting more efficient recruitment of RNApol2 and Integrator as a potential explanation for their higher expression (Fig. 6c). We conclude that broad promoters have higher and broader signals for Mediator that is closely tracked by all its tested interacting factors.

**Tcf4 regulates genes with SEs and broad H3K4me3 promoters.** Tcf4 showed the highest overlap with Mediator at enhancers and SEs of the tested Mediator-interacting TFs (Fig. 4a) prompting us to further investigate a possible role of Tcf4 in regulating genes near SEs. We find that Tcf4 content followed Mediator content at enhancers and SEs (Fig. 7a). To test to what extent Tcf4 regulates genes with or without SEs and/or broad H3K4me3 promoters, we used our RNA-seq dataset from RNA isolated 44 h after Tcf4 knock-down or control knock-down in NSCs[28]. We found that Tcf4 depletion downregulates nearly two-thirds of all SE + Broad genes (Fig. 7b) and also has the strongest downregulating effect on SE-containing genes (Fig. 7c). Genes without SEs, either

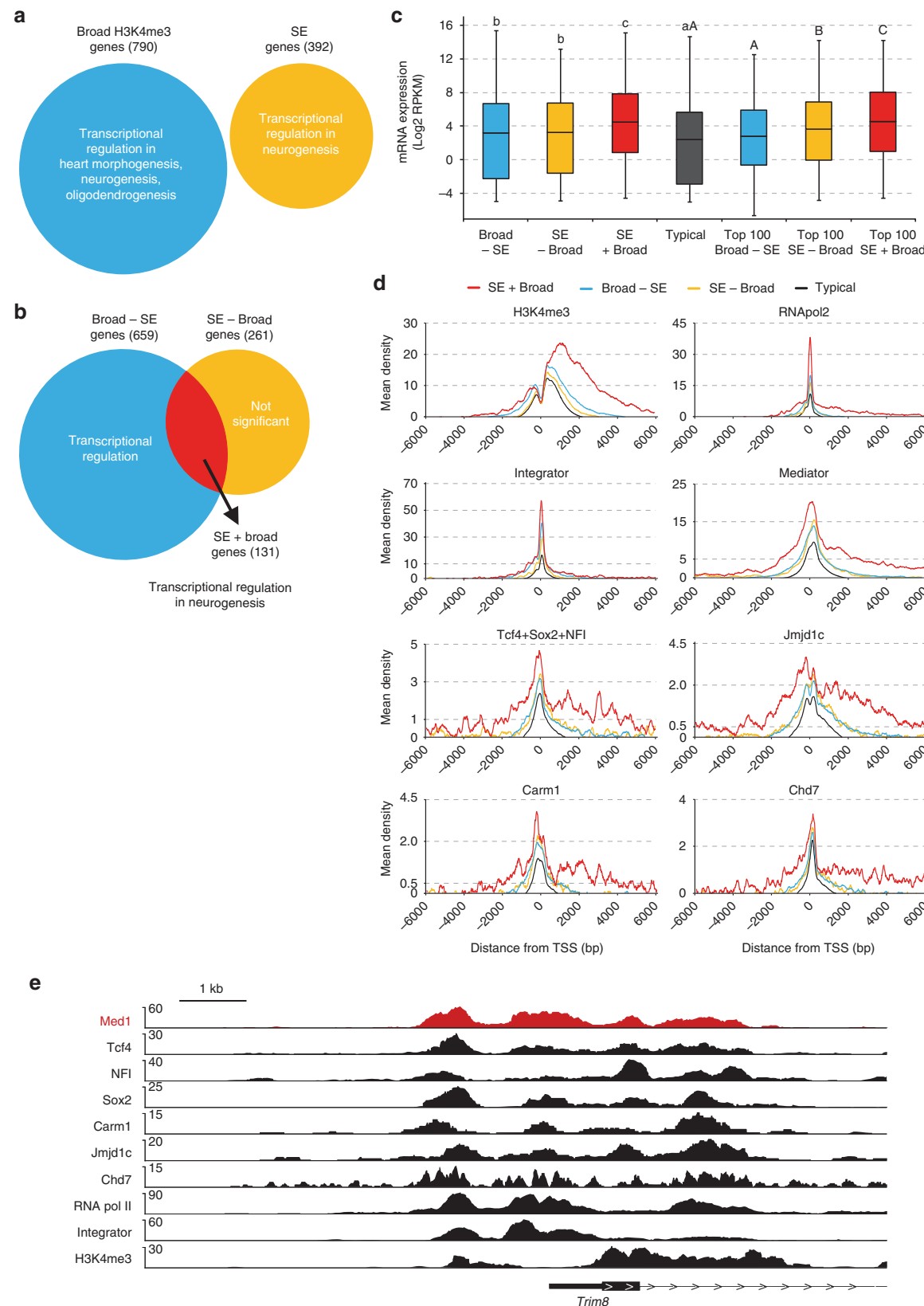

Broad-SE genes or genes with typical enhancers, are significantly less affected by Tcf4 depletion (Fig. 7c). This suggests that Tcf4 predominantly regulates genes via SEs. Indeed, Tcf4 is present on nearly all SEs of SE + Broad and SE-Broad genes (Fig. 7d). Tcf4-bound and activated SE + Broad genes include 15 transcription factor genes (Fig. 7e) of which *Bahcc1*, *Hes1, Myc*, *Nfib*, *Sall1* and *Sall3, Thra* and *Npas3* encode known regulators of neural progenitors and/or neurogenesis[31,34–39]. Tcf4 protein has protein–protein interactions in NSCs with 6 TFs that are part of this set of Tcf4-activated TF genes, including Nfib and Olig2[28]

**Fig. 6** Mediator complex and its interactors at promoters. **a** Predominant Gene Ontology terms for genes with broad H3k4me3 promoters and for active genes nearest to SEs (SE genes) in NSCs. Numbers of genes in each category are indicated between brackets. **b** Overlap of genes with broad H3K4me3 promoters and SE genes in NSCs. Venn diagram with the two categories of genes, their overlap and their predominant Gene Ontology terms is shown. Numbers of genes in each category are indicated between brackets. **c** Distribution of mRNA levels in NSCs of the different categories of active genes. Box plots based on RNA-seq triplicate data are shown. Broad-SE, broad H3K4me3 promoter genes not nearest to SE. SE-Broad, SE genes without broad H3K4me3 promoter. SE + Broad, SE genes with broad H3K4me3 promoter. Typical, genes nearest to a typical enhancer but not nearest to an SE and without a broad H3K4me3 promoter. mRNA levels of all genes and top 100 genes within each category are shown. Statistically significant differences between groups are indicated as separate letters above the box plots, as assessed by Student t-tests comparing all gene subsets (lower case letters) or top 100 subsets (upper case letters). $p < 0.001$ except for B, $p < 0.05$. If the letters are the same, the difference between these groups is not significant. Source data are provided as a Source Data file. **d** ChIP-seq density plots around promoters of the different categories of genes for the indicated factors and histone modifications. Mean ChIP-seq density (y-axis) and distance to TSS (x-axis) are shown. **e** Overlap of binding sites of Med1 with binding sites of Mediator interactors at the *Trim8* broad H3K4me3 promotor area in NSCs. ChIP-seq tracks for the indicated proteins and histone modifications at the *Trim8* gene are shown. Range of reads per million per base pair is indicated on the y-axis. Scale bar is indicated

(Fig. 7e). This allows for a potential feed-forward circuit (Fig. 7e) where Tcf4 maintains the expression of its own co-factors, which then subsequently may aid Tcf4 in the regulation of other target genes and its own expression. In line with this possibility, NFI and Olig2 colocalize with Tcf4 and Mediator on SEs in all 15 TF genes, for example at the *Olig2* gene (Fig. 7f), the *Sall3* gene and the *Notch1* gene (Supplementary Fig. 2). Tcf4, Mediator, NFI and Olig2 also colocalize at the SE in the *Tcf4* gene itself (Fig. 7g). *Nfib* expression has the second-best spatial-temporal correlation (0.56 Pearson coefficient) with *Tcf4* expression in pre-natal development of the mouse brain (out of 1104 genes)[40] and the second-best spatial-temporal correlation (0.90 Pearson coefficient) with *TCF4* in pre-natal human brain development (out of 19700 genes)[41], suggesting that a Tcf4-Nfib co-regulatory partnership could be widespread in mammalian brain development.

## Discussion
We have expanded the protein–protein interaction network of the Mediator complex with many proteins and complexes that reside at enhancers, super enhancers or promoters and thereby established the potential of the Mediator complex as a major interaction hub at enhancer-promoter assemblies. Mediator binds to enhancers and promoters in close proximity to many other proteins. We believed that chromatin-independent protein–protein interactions of purified Mediator complex, as identified by their detection by mass spectrometry, would be the best indicator of its recruitment capacity. Despite our stringent criteria, 20 years of research on the Mediator complex since its discovery by several labs[25,42–44] and progressing high throughput interaction studies[22,45], we find that 75 of our 95 identified Mediator interactions have not been, to the best of our knowledge, previously characterized.

Identified Mediator interactors can be broadly divided into DNA sequence-independent proteins, mostly chromatin modifiers, and sequence-specific transcription factors. The latter category of Mediator interactors would represent potential Mediator-recruitment factors. Indeed, NF-kappaB subunit RelA, one of the two known Mediator interactors among the 16 identified transcription factors, recruits Mediator to activate transcription[46]. Whereas Mediator-interacting transcription factors would be more specific for NSCs (see next paragraph), the Mediator-interacting chromatin modifiers and other proteins are mostly ubiquitously expressed and would have general relevance for transcriptional regulation. Supporting this suggestion, our Mediator interactor screen discovered two major enhancer-binding proteins. We observed and independently confirmed interactions between Mediator and arginine methylase Carm1 and putative H3K9 demethylase Jmjd1c. Carm1 is a highly studied enzyme and best known in transcriptional regulation as a co-activator of nuclear receptors and NF-kappaB and was shown to

act at individual promoters[47,48]. We find that Carm1 is a genome-wide enhancer-binding protein in NSCs that closely colocalizes with Mediator. Jmjd1c was identified as a co-activator of the tumor-inducing fusion gene AML1-ETO and shown to be recruited by AML1-ETO to target gene promoters where it lowers the levels of the repressive mark H3K9me2[49]. We show that Jmjd1c marks enhancers genome-wide in NSCs, together with Mediator, where it may perform a similar enzymatic role to maintain enhancer activity.

A recent analysis[8] showed that chromatin modifiers Brd4, Ep300, Crebbp, Chd7, SWI-SNF complex, LSD1 complex, Cohesin complex and NuRD complex colocalize with Mediator at enhancers and have an increased binding density at SEs, similar to the Mediator complex. With the exception of Brd4, we find all the above-mentioned chromatin modifiers as Mediator inter-actors, which may suggest that Mediator interaction aids in their recruitment to enhancers and SEs. The apparent correlation of having protein–protein interactions with Mediator and coloca-lising with Mediator on the genome would predict that other observed Mediator interactors of unknown genomic location also reside at enhancers or promoters. This remains to be tested.

We performed Mediator ChIP-seq to identify SEs in NSCs. We find that Mediator-defined SEs in NSCs have as their most fre-quent TF motifs E-box, NFI and SOX, similar to NSC enhancers in general[27]. Nfia, Nfib, Sox2, Tcf4 and Tcf12, which can bind one of these motifs, are among the small set of 16 TFs that we identified as Mediator interactors. This shows a remarkable synchrony between Mediator-binding TFs and prominent enhancer motifs in NSCs. Our identified Mediator-binding TFs are not the highest expressed TFs in NSCs, suggesting that they have a higher binding affinity for Mediator than other TFs. The above set of TFs may therefore define enhancers and SEs in NSCs by having high affinity for Mediator and thereby being effective at recruiting Mediator and its interactors to its binding sites. Accordingly, we find that Tcf4 and Sox2 are required for optimal Mediator recruitment to some of the tested genomic sites where the three factors have overlapping binding. This would suggest Mediator affinity as an important organizing feature in estab-lishing the enhancer landscape in a given cell type. Indeed, the sum of the binding sites of Tcf4, Sox2 and Nfi represents nearly all Mediator-binding sites at enhancers, and outside promoters in general, and can therefore explain genome-wide Mediator recruitment outside promoters in NSCs.

Relative promoter occupancy of Mediator has not been ana-lysed genome-wide in higher eukaryotes, to our knowledge. We find that Mediator has higher and especially broader binding signals at promoters with a broad H3K4me3 signal, a class of promoters that was recently discovered[31,32]. Tcf4, Sox2 and Nfi show relatively weak occupancy at promoters in general. How-ever, their binding is enhanced at broad promoters and Mediator

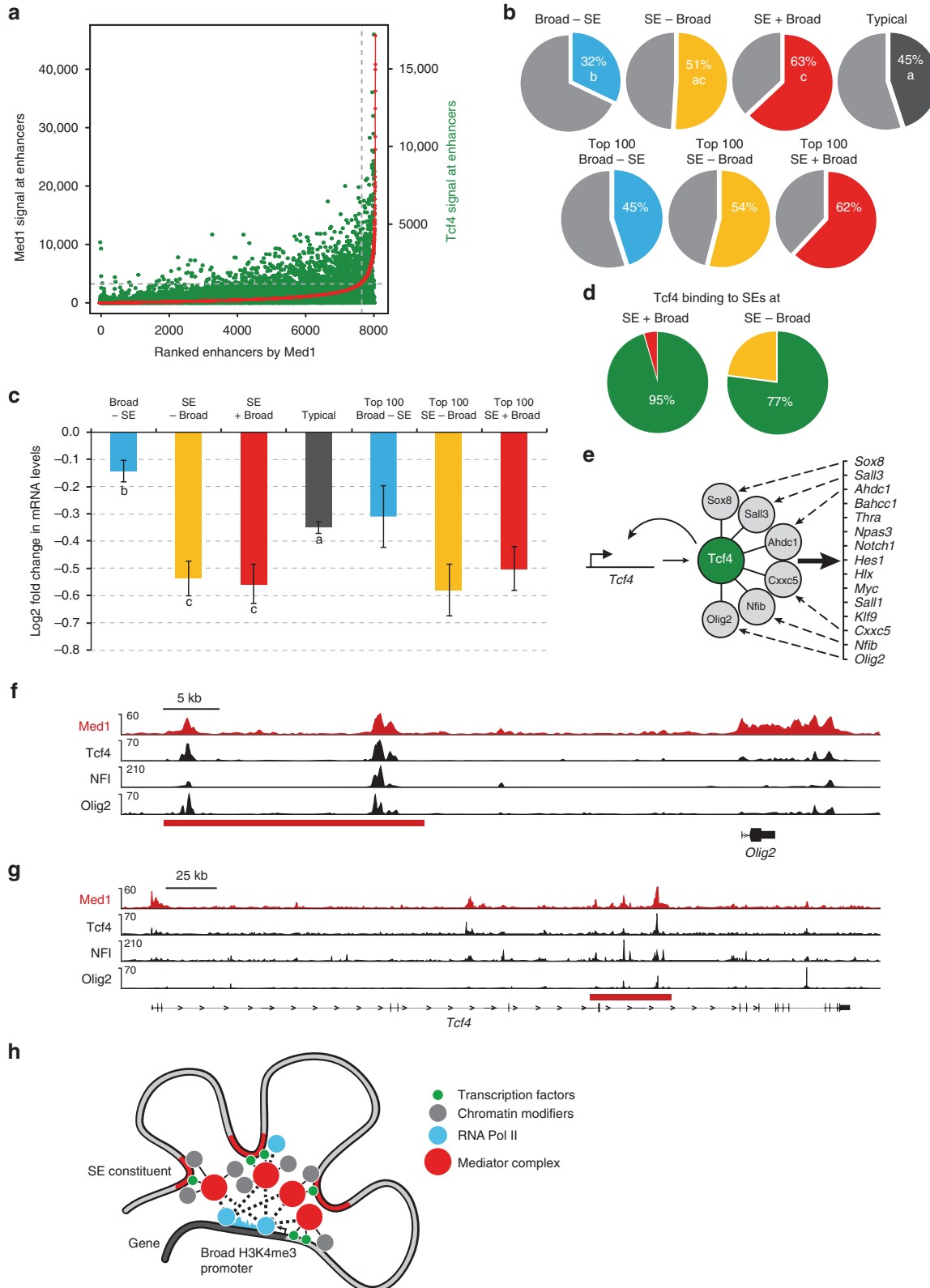

follows closely their binding pattern in our genome-wide plots, as well as at individual broad promoters. We find that Mediator-interacting chromatin modifiers, such as Carm1 and Jmjd1c, also track Mediator binding at promoters. Relative enrichment of transcription factors at broad promoters was observed before in different cell types[31,32]. Our results suggest that broad promoters may act like proximal enhancers in recruiting TFs, which in turn

can recruit Mediator and its interactors. The close resemblance of the Mediator genome-wide binding sites with the binding sites of its interacting TFs is highly suggestive of Mediator recruitment by these TFs.

We find that Tcf4 preferentially regulates SE-containing genes in NSCs, including a set of neurogenic transcription factor genes that have SEs and broad promoters. Intriguingly, we find that a

**Fig. 7** Tcf4 regulates neurogenic transcription factor genes with super enhancers and broad H3K4me3 promoters. **a** Tcf4 signal at enhancers ranked by Med1 content. Tcf4 ChIP-seq read content is in green, enhancers ranked by Med1 ChIP-seq read content is in red. **b** Percentages of downregulated genes in the different categories upon Tcf4 knock-down in NSCs. Percentages of down-regulated genes in all genes and top 100 genes within each category are shown. Statistically significant differences between groups are indicated as separate letters in the pie charts, $p < 0.001$ as assessed by Student $t$-tests. **c** Changes in mRNA levels of the different categories of genes upon Tcf4 knock-down in NSCs. Log2 fold change, based on RNA-seq data, is shown. Error bars indicate S.e.m., based on the RNA-seq triplicates. Statistically significant differences between groups are indicated as separate letters below the box plots, $p < 0.001$ as assessed by Student $t$-tests. Source data are provided as a Source Data file. **d** Percentage of Tcf4-bound SEs in SE + Broad genes or SE-Broad genes in NSCs. SEs nearest to SE + Broad genes or SE-Broad genes with or without significant Tcf4 binding sites, as determined by ChIP-seq, were counted. **e** Model of Tcf4-driven feed-forward transcriptional circuit of SE + Broad TF genes in NSCs. Fifteen15 SE + Broad TF genes bound at their SE and activated by Tcf4 are indicated. Tcf4 also binds its own SE. TF proteins encoded by six target genes also interact with Tcf4 protein and may aid in transcriptional regulation by Tcf4. **f, g** Overlap of binding sites of Tcf4 and Med1 with Tcf4-interactors Olig2 and NFI at the *Olig2* gene (**f**) or *Tcf4* gene (**g**) in NSCs. ChIP-seq tracks for the indicated proteins are shown. SE is indicated with a red bar. Range of reads per million per base pair is indicated on the y-axis. Scale bar is indicated. **h** Model of SE-Broad H3K4me3 promoter assemblies. TFs at SE constituents and the Broad H3K4me3 promoter recruit high levels of Mediator complex into SE-Broad assemblies. In turn, Mediator recruits high levels of protein–protein interaction partners such as the RNApol2 complex, Integrator, and chromatin modifiers. This would result in efficient pause-release of RNApol2 and high but TF-regulated levels of transcription

number of the TFs encoded by these genes have protein–protein interactions with the Tcf4 protein. Some of these Tcf4-interacting TFs colocalize with Tcf4 at SEs in this set of target genes, as well as on the Tcf4 gene itself, suggesting a feed-forward circuit that maintains the expression of these TFs in NSCs. Feed-forward circuits of key TFs in embryonic stem cells (ESCs), such as Oct4, Sox2, Nanog, Esrrb and Klf4, were shown to regulate pluripotency and follow the same above criteria[7,50,51]. Analogous to the ESC TF circuit, many of the TFs in our NSC circuit are essential for NSC self-renewal or their neuronal differentiation capacity. Together, this suggests that we have uncovered a TF circuit that would be central to the regulation of NSC identity. *TCF4* heterozygosity in humans leads to Pitt Hopkins syndrome with severe intellectual disability[52,53], whereas SNPs in the *TCF4* locus are the most significant schizophrenia risk SNPs to date[54]. These genetic data suggest that TCF4 plays an important role in brain development and needs to be tightly regulated to prevent neurodevelopmental disease. Our TF circuit may facilitate this regulation.

Mediator complex binding signal was used as one parameter to postulate SEs[7], which were subsequently shown to regulate cell-identity genes and oncogenes in many cell types[7–9]. More recently, promoters with a broad H3K4me3 domain were postulated to regulate cell-identity genes[31,32]. As was shown before in other cell types[31,32], we find that SE genes and Broad genes partially overlap in NSCs. However, we show that the link to neurogenic transcriptional regulators in SE genes and Broad genes in NSCs is derived from neurogenic transcriptional regulator genes in the overlap of both categories; genes that have both SEs and broad promoters. This suggests that, at least in NSCs, SE + Broad genes represent a special category of genes that is strongly linked to cell identity. These SE + Broad genes have high recruitment of Mediator at their SEs (by definition) and we find that they also recruit high levels of Mediator to their promoters. Increased promoter levels of Mediator are also observed at broad promoters without surrounding SEs may therefore be recruited by Mediator-interacting TFs, which we also find enriched at broad promoters. SEs were recently shown to have increased 3D interactions with broad promoters, as compared to typical promoters[33]. We find that SE + Broad genes in NSCs are the category of genes with highest levels of RNApol2 and Integrator at their promoters. Integrator complex associates with RNApol2 and plays an important role in the transcription-initiation and pause-release of RNApol2[30]. The efficient recruitment of RNApol2 and Integrator at SE + Broad genes thereby provides an explanation for our observation that this category of genes has the highest expression in NSCs.

All together this fits into a model (Fig. 7h) where Mediator is recruited by Mediator-interacting TFs to both SEs and Broad

promoters. These elements then form relatively stable enhancer-promoter assemblies that have high local concentrations of Mediator and its co-recruited protein–protein interaction partners, including RNApol2, Integrator and chromatin modifiers. Such assemblies would provide an optimal environment for the efficient pause-release of high quantities of RNApol2 and thereby combine the high transcriptional consistency and the high transcriptional efficiency that have been shown for broad promoters and SE genes, respectively[7,8,31]. SE-broad promoter assemblies and our identified Mediator interactions could provide ideal building blocks for the phase-separated complexes that have been recently proposed to drive robust transcription of cell-identity genes in mammals[14].

## Methods

**Purification of the Mediator complex from neural stem cells.** NS-5 neural stem cells (NSCs) were derived from 46 C embryonic stem cells[55] and cultured on N2B27 medium (Stem Cell Sciences) supplemented with EGF and FGF (both from Peprotech)[56] and regularly tested for mycoplasma contamination. Essentially all our NSCs express NSC markers Sox2 and Nestin (Supplementary Fig. 3a and b). NSC lines with stable expression of C-terminally FLAG-tagged Med15 were created by electroporation with pCAG promoter-driven plasmids containing Med15 cDNA and puromycin selection for individual clones with moderate expression of the tagged proteins, as compared to endogenous levels[20,28]. Nuclear extract was prepared from NSCs expressing FLAG-Med15 and from control NSCs by the classical Dignam protocol[21] and FLAG-tagged Mediator complex was purified from 1.5 ml nuclear extract, equivalent to $2 \times 10^8$ NSCs, by FLAG-affinity purification, and analyzed by mass spectrometry, as described[19,20]. In brief, nuclear extracts were dialyzed to 20 mM Hepes pH7.6, 0.2 mM EDTA, 1.5 mM MgCl2, 100 mM KCl, 20% glycerol (buffer C-100). Eighty microlitre of anti-Flag M2 agarose beads (Sigma) equilibrated in buffer C-100 were added to 1.5 ml of nuclear extract and incubated for 3 h at 4 °C in the presence of Benzonase (Novagen). Beads were washed five times with buffer C-100 containing 0.02% NP-40 (C-100*) and bound proteins were subsequently eluted at 4 °C with buffer C-100* containing 0.2 mg/ml Flag-tripeptide (Sigma). Elutions were TCA precipitated, separated on a 10% NuPAGE Bis-Tris gel (Invitrogen) and stained with Colloidal Coomassie (Biorad) according to manufacturer's instructions. Gel lanes were cut and subjected to in-gel digestion with trypsin (Promega). Nano-LC-MS/MS was performed on an 11 series capillary LC system (Agilent Technologies) coupled to an LTQ mass spectrometer (Thermo). Peptide spectra from purified Mediator samples or control sample were searched against UniProt release 2012-11 for protein identification using MASCOT.

Mediator complex purifications were performed from nuclear extract with Benzonase (150 U per ml nuclear extract) added or Ethidium bromide (50 μg per ml) added at the start of the 3-h incubation period of the anti-FLAG antibody beads with the nuclear extract. Alternatively, Mediator complex purification was performed from untreated nuclear extract. In one experiment, Mediator complex purifications were performed from nuclear extracts treated with Benzonase, Ethidium bromide or untreated nuclear extract, together with a control purification from nuclear extract from control NSCs. In a second, independent, experiment, Mediator complex was purified from nuclear extract treated with Benzonase, together with a control purification. Control purifications were from nuclear extract treated with benzonase. All purifications are shown in Supplementary Data 1. An uncropped image of the DNA gel of Fig. 1b can be found in Supplementary Fig. 4a.

Initial inclusion criteria for Mediator-interacting proteins are as described[19]; (1) A minimal Mascot score of 50, (2) At least five-fold enrichment by emPAI score in

the Mediator purified sample over the control sample. emPAI score is an estimate of the quantity of the identified protein in the purified protein sample, based on the number of peptide spectra identified by MS, normalized for the number of peptides that theoretically should be identifiable for that protein[57]. (3) At least three-fold enrichment by Mascot score in the mediator purified sample over the control sample. (4) Cytoskeletal and cytoplasmic proteins (Uniprot) were removed. Of note, of the 96 identified Mediator complex interactors, only 12 are also detected in any of the two control samples (Supplementary Data 1).

Subsequently, recorded Mediator interactors cannot be two-fold lower or more in emPAI score in the Mediator complex purification from purifications from nuclear extracts treated with Benzonase or Ethidium bromide, as compared to a parallel Mediator complex purification from untreated nuclear extract. Finally, Mediator interaction partners are only included in the final list (Supplementary Data 1) if they are specifically present in all four Mediator complex purifications. Mediator interaction partners were defined as novel if they did not appear as identified by Affinity Capture or Reconstituted Complex in BioGRID, the most comprehensive protein–protein interaction database[58]. Interaction network graphics were made with Cytoscape[59]. Thickness of the edges in the interaction network (Fig. 1c) gives an indication of the relative molar protein quantity (based on emPAI score) in purified Mediator complex samples with 4 categories of thickness; emPAI > 1.5, thickest edge, 0.75 < emPAI ≤ 1.5, one but thickest edge, 0.25 < emPAI ≤ 0.75, one but thinnest edge, emPAI < 0.25, thinnest edge.

**Immunoprecipitations**. Immunoprecipitation of Med12 was performed from 1.5 ml of NSC nuclear extract using 15 μg Med12 antibody (Bethyl Laboratories #A300-774A) crosslinked by dimethyl pimelimidate (Sigma) to 50 μl (pellet volume) of protein G sepharose beads (GE17-0618-01, Sigma), as described[19]. Med12 antibody beads were blocked with 0.1 mg/ml insulin (Sigma), 0.2 mg/ml chicken egg albumin (Sigma), 1% fish skin gelatin (Sigma) and added to 1.5 ml of nuclear extract with or without 225 units of Benzonase (Novagen) and rotated for 3 h at 4 °C in no-stick microcentrifuge tubes (Alpha Laboratories), washed five times with 1 ml of C-100* buffer (20 mM Hepes pH 7.6, 0.2 mM EDTA, 1.5 mM MgCl2, 100 mM KCl, 20% glycerol) at 4 °C and proteins eluted from the beads by 5 min at 95 °C in 50 μl SDS-loading dye. Eluted proteins were separated by poly-acrylamide gelelectrophoresis and analyzed by mass spectrometry, as described above. Inclusion criteria for endogenous Mediator interactors (Supplementary Data 2) are as for the FLAG-Mediator purification, except for the requirements on emPAI score ratio in benzonase versus no benzonase samples. Immunoprecipitations of Jmjd1c or Carm1 were performed from 1 ml of NSC nuclear extract treated with benzonase and using 10 μg of Jmjd1c antibody (Merck Millipore #17-10262), or 10 μg Carm1 antibody (Cell Signaling Technology #12495) or 10 μg of control rabbit IgG (Santa Cruz #sc-2027) were as described above but without mass spectrometry analyses. Resulting western blots were performed with PBS 0.1% Tween solutions, blocking in 5% Fat-free milk proteins and probing with Jmjd1c antibody (Merck Millipore #17-10262, 1:1000), Med12 antibody (Bethyl Laboratories #A300-774A, 1:1000) and Donkey anti-rabbit HRP-conjugates (Sigma #GENA934, 1:2500). Uncropped images of the westerns are found in Supplementary Fig. 4b–e.

**Mediator-interacting TFs**. References for function in neural development. Yy1[60], Nfia[61], Sox2[62], Nfib[63], Sall3[31], Tcf12[64], Znf24[65], Rela[66], Tcf4[67,68]. TF mRNA levels in our NSCs are from our RNA-seq data on our wild-type NSCs[28].

**ChIP-seq**. We adapted protocols previously described[7,28]. $1.5 \times 10^8$ NSCs were used per chromatin immunoprecipitation (ChIP). Cells were collected in 1xPBS and crosslinked first with 2 mM disuccinimidyl glutarate (Thermo Fisher Scientific, Waltham, MA, USA) solution for 45 min and then 1% formaldehyde solution for 15 min at room temperature. Cells were washed twice with 1X PBS and flash frozen in liquid nitrogen. Chromatin was prepared for sonication with 20 mM Tris-HCl pH8, 150 mM NaCl, 2 mM EDTA, 0.1% SDS, 1% Triton X-100. We used 15 cycles of 30 s ON, 30 s OFF on a Bioruptor Pico sonication device (Diagenode Cat# B01060001) to shear chromatin to 150–200 bp fragments. The resulting 300 μg of chromatin extract was incubated overnight at 4 °C with 100ul of Dynal Protein G magnetic beads that had been pre-incubated with 10 μg of the appropriate antibody. We used the following antibodies: Med1 (Bethyl Labs #A300-793A), Carm1 (Cell Signaling Technology #12495), Jmjd1c (Merck Millipore #17-10262), IgG (Normal Rabbit IgG: Santa Cruz #sc-2027). Beads were washed 1X with the sonication buffer, 1X with 20 mM Tris-HCl pH8, 500 mM NaCl, 2 mM EDTA, 0.1% SDS, 1%Triton X-100, 1X with 10 mM Tris-HCl pH8, 250 mM LiCl, 2 mM EDTA, 1% NP40 and 1X with TE containing 50 mM NaCl. Bound complexes were eluted from the beads in 50 mM Tris-HCl, pH 8.0, 10 mM EDTA and 1% SDS by heating at 65 °C for 1 hr with occasional vortexing and crosslinking was reversed by overnight incubation at 65 °C. ChIP-seq sample preparation and sequencing on Illumina GAII or HiSeq2500 (San Diego, CA, USA) platforms was performed at the Erasmus MC Center for Biomics, according to manufacturer's instructions.

**ChIP in combination with knock-down**. For Tcf4 knock-down, 36 transfections of each 3.5 μg of pSuper-puro-Tcf4-shRNA#1[28] into $3.5 \times 10^6$ NSCs (in total $126 \times 10^6$ NSCs were transfected) were performed using program A-33 on the Amaxa

nucleofector I, kit Cell Line Nucleofector™ Kit V (Lonza, catalog # VVCA-1003) and plated on 36 6-cm dishes. As control, pSuper-puro-Control-shRNA transfections were performed with the same set-up. Control (scrambled control sequence from Dharmacon) shRNA sequence: GGTGAGCTTCATGAGGATG. Selection was started 20 h after transfection with 2 μg/ml Puromycin and NSCs were collected after 28 h of selection (48 h after transfection). For Sox2 knock-down, 36 transfection of pSuper-puro-Sox2-shRNA#1[20] into NSCs were performed using the same set-up as for Tcf4. As control, pSuper-puro-Control-shRNA transfections were performed with the same set-up. Selection was started 18 h after transfection with 2 μg/ml Puromycin and cells were harvested after 28 h of selection (46 h after transfection). Before crosslinking the NSCs, one 6-cm dish for each shRNA construct was collected to verify the gene expression of Tcf4 and Sox2 by qRT-PCR. RNA was isolated using the GenElute™ Mammalian Total RNA Miniprep Kit (Sigma, RTN350-1KT) and reverted to cDNA using RevertAid First Strand cDNA Synthesis Kit (Fermentas, K1621).

ChIP was performed as described above on 30 μg of chromatin per condition with 30 μl of Dyna Protein G beads pre-incubated with either 3 μg of Med12 antibody (Bethyl Laboratories #A300-774A) or 3 μg rabbit IgG (Santa Cruz #sc-2027) for control ChIPs. Tcf4 knock-down ChIPs, Sox2 knock-down ChIPs and their respective control ChIPs were performed in biological duplicate. RT-PCR was performed on genomic targets indicated with their distance from the TSS of the nearest gene with—indicating upstream of the TSS and +indicating downstream of the TSS. Supplementary Table 2 lists all used primers.

**Immunocytochemistry**. Neural stem cells were grown on poly-D-lysine (0.5 mg/ml, Sigma–Aldrich) coated glass coverslips and fixed in 4% Paraformaldehyde for 10 min at room temperature. After fixation, the cells were washed with PBS+ (0.5% Bovine Serum Albumin and 0.15% Glycine in PBS) and permeabilized with PBS-0.1% Triton X-100. Subsequently, slides were incubated with the primary antibodies (Rabbit anti-Nestin, Biolegend® 839801, 1:200 dilution, and goat anti-Sox2, Santa-Cruz Biotechnology sc-17320, 1:200 dilution) diluted in PBS+ for 2 h at room temperature. After subsequent washes with PBS-0.1% Triton X-100 and with PBS+, the coverslips were incubated with secondary antibodies (Donkey anti-Rabbit Alexafluor 488 and Donkey anti-Goat Alexafluor 594) diluted in PBS+ and washed with PBS-0.1% Triton X-100 and PBS+. The samples were mounted with MOWIOL (#324590, Sigma–Aldrich) and nuclei were stained with DAPI (Vector laboratories). Images were captured using a Leica DM4000 B fluorescent microscope and image processing was performed using FIJI (ImageJ). Sox2-positive NSCs or Nestin-positive NSCs were counted.

**Genomic data analyses**. All ChIP-seq datasets were mapped to the mouse mm9 reference genome using Bowtie v0.12.7[69], where we used a seed length of 36 in which we allowed a maximum of two mismatches. If a read had multiple alignments only the best matching read was reported. ChIP-seq datasets with multiple replicates were merged. Duplicated reads were removed. MACS46 v1.4.2 was used for peak calling using default settings, using IgG ChIP-seq as background control for our Med1, Carm1, Jmjd1c, Tcf4, Olig2 and Chd7 ChIP-seq data. For external ChIP-seq datasets either IgG ChIP-seq or sequenced chromatin input was used as background control. For histone modifications we used HOMER findPeaks[70] using -region -size 1000 -minDist 2500 parameters. Genomic datasets that are generated and/or used in this study are summarized in Supplementary Table 1.

Enhancers in mouse NSCs were defined by recalling Ep300 and H3K27ac peaks using HOMER, function REGION, and using Bedtools[71] to generate overlaps between Ep300 peaks and H3K27ac peaks. SEs were identified using the ROSE algorithm[7]. ROSE stitches together enhancers that are within 12.5 kb of each other and do not overlap with a window of 1 kb on either side of a TSS and ranks such combined enhancers by their total Med1 ChIP-seq signal[7]. Four hundred forty-five super enhancers were identified and the rest were assigned as typical enhancers. Plotting was performed using *hockey* function in R We used the already described list of mouse NSCs broad H3K4me3 promoters[31].

For mRNA levels in our mouse NSCs, we used our published RNA sequencing dataset[28] consisting of three replicates to calculate the mean mRNA expression levels. Super enhancer (SE) genes and typical enhancer genes are defined as the closest active gene, RKPM > 0.5 in our NSC RNA-seq data[28], to an SE or a typical enhancer, respectively.

Motif analyses were performed using HOMER[70] and selecting the most frequent motifs found at Med1 binding sites at SE constituents and typical enhancers.

For genome-wide binding site overlaps, we used the 5000 most significant binding sites for each factor to determine the percentage of overlap between two factors. Two binding sites were considered overlapping if their summits were within 200 bp. Promoters were defined as the regions within 1.5 kb of a transcription start site (TSS). Top 5000 peaks from Mediator and its interactors were separated in the TSS, non TSS, typical enhancer and super enhancer categories and the percentage of overlap recalculated for each subset.

Generation of histograms documenting ChIP-seq signal density at specific sets of promoters in the NSC genome was performed by HOMER annotatePeaks with 10 bp bins and 12,000 bp around the TSS. By default, HOMER normalizes the output histogram such that the resulting units are per bp per peak, on top of the standard total mapped tag normalization of 10 million tags. For each promoter,

directionality was extracted from TSS annotation and each subset was split between plus or minus strand. Subsequently, split lists were then remerged taking into account directionality to finally calculate the ChIP-signal density values. Enhancer-annotated expressed genes not present on the super enhancer gene list or the broad H3K4me3 promoter gene list were used as typical genes.

Gene Ontology analyses on the different gene categories were performed using DAVID version 6.7[72] using default categories. SE genes are defined as the closest active gene, RKPM > 0.5 in our NSC RNA-seq data[28], to an SE. Broad H3K4me3 genes are defined as having the top 5% broadest H3K4me3 domains[31]. Additionally, we performed GO ontology biological process analysis of the transcription regulators found in each subset. Benjamini-corrected p-value was used for ranking Gene Ontology terms.

Top 100 SE + Broad genes are genes with a broad promoter and from that category the 100 genes that are the nearest gene to the highest SEs (i.e. one gene per SE), as ranked by Mediator signal. Top 100 SE-Broad genes are genes without a broad promoter and from that category the 100 genes that are the nearest gene to the highest SEs, as ranked by Mediator signal. Top 100 Broad-SE genes are genes that not the nearest gene to an SE and from that category the 100 genes ranked by the broadest H3K4me3 signal at their promoter, from an already described list of mouse NSC broad H3K4me3 promoters[31].

Tcf4-regulated genes were derived from an RNA-seq. experiment performed in triplicate 48 h after Tcf4-shRNA-transfection or control shRNA-transfection in mouse NSCs[28]. Gene expression values with significant triplicates were assigned to the different subsets. The effect of Tcf4 knock-down was indicated by plotting the mean fold change vs the scrambled shRNA condition for each subset.

**Reporting summary**. Further information on research design is available in the Nature Research Reporting Summary linked to this article.

## Data availability
The genomic data reported in this paper were submitted to GEO database under accession number: GSE109043. The mass spectrometry proteomics data have been deposited to the ProteomeXchange Consortium via the PRIDE partner[73] repository with the dataset identifier PXD013546. Fig. 1c and Supplementary Data 1–3 have associated proteomic data. Figs. 2d, 3, 4, 6 and 7, Supplementary Figs. 1 and 2, Supplementary Data 4 and 5 have associated genomic data. The source data underlying Figs. 1b, 2a, b, 3a, c, 5a–d, 6c, 7c are provided as a Source Data file. All other relevant data supporting the key findings of this study are available within the article and its Supplementary Information files or from the corresponding author upon reasonable request. A reporting summary for this Article is available as a Supplementary Information file.

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

## Acknowledgements

We thank Christel Kockx for technical assistance. M.Q. was supported by a graduate programme grant (nr. 022.004.002) from the the Netherlands Organisation for Scientific Research (NWO). M.R.D was supported by the DevRepair (P7/07) IAP-VII network, J.H.B. was supported by an ALW-open program grant (No 821.02.004) from NWO and R.A.P. and D.L.C.v.d.B. by a grant from the Dutch government to the Netherlands Institute for Regenerative Medicine (NIRM, grant No. FES0908). D.H.D. and J.D. were funded by The Netherlands Proteomics Centre (Project Number 184.032.201), financed by NWO.

## Author contributions

M.Q. performed nearly all experiments and designed and performed most of the bioinformatic analyses. L.M. performed the Med12 ChIP upon Tcf4-Sox2 knock-down experiments and NSC antibody stainings. M.R.D. assisted in NSC transfections and NSC nuclear extract preparation and other experiments. D.H.W.D. and J.D. performed the mass spectrometry analyses. D.L.C.v.d.B. performed plasmid constructions and preliminary experiments. Z.O. and W.F.J. van IJ. performed labeling and Illumina sequencing of ChIP material. M.F. designed, performed and supervised part of the bioinformatic analyses and correspondence on computational biology should be addressed to M.F.; m.fornerod@erasmusmc.nl. R.A.P. conceived the study and designed experiments. and wrote the manuscript with help from co-authors.

## Additional information

**Competing Interests:** The authors declare no competing interests.

