## [Peer Review File · Nature Communications]

Reviewers' comments:

Reviewer #1 (Remarks to the Author):

The Quevedo et al. investigated the Mediator interaction partners that organize the transcriptional network in defining the neural stem cell identity. The mass spec data is impressive, and integrative sequencing and bioinformatic analysis are kind of comprehensive. Here are a few specific comments:

1. One would wonder how similar is between the mouse neural stem cell line with the real neural stem cells in vivo. For cell line derived from iPSCs, the heterogeneity in cell population is also a problem. Since all the conclusions of this study came from the experiments using this cell line, these aforementioned questions should be addressed by comparing the transcriptome between this cell line with NSCs derived from mouse brain. Single cell sequencing or immunostaining experiments should be done to address the heterogeneity issue.
2. For establishing the Flag-tagged MED15 NSCs, it should be checked to see how much of MED15 is incorporated into the endogenous Mediator complex using Co-IP or gel-filtration experiment. The exogenous expressed subunit of protein complex might be hard to integrate into the complex in vivo. Too much free-form of Flag-MED15 could skew the Co-IP plus mass spec results.
3. It is interesting that the Mediator associated partners in NSCs include RNA binding factors. Mediator was reported to be involved in mRNA processing, which might be relevant and should be discussed.
4. Through this investigation, a Mediator interaction network of enhancer- and promoter- binding proteins, however, all these protein partners are not necessarily in direct contact with the Mediator complex. Association rather than interaction could be most of the cases, however, the authors have used "interaction" throughout the whole manuscript.

Reviewer #2 (Remarks to the Author):

I read with great interest the manuscript by Quevedo et al. entitled "Mediator complex interaction partners organize the transcriptional network that defines neural stem cells". In this manuscript the authors use a proteomics approach to systematically and globally identify proteins interacting with the Mediator complex in neural stem cells. As a result they identified a large number of previously unknown interactors, including several chromatin modifiers and transcription factors. Moreover, using both publically available as well as new ChIP-seq data, the authors show that several of these newly identified interactors co-localize with Mediator at enhancers in general and at super-enhancers in particular. Based on these observations, the authors suggest that some of the identified transcription factors (e.g. TCF4) might recruit Mediator to enhancers and, thus, play major roles in establishing the neural stem cell regulatory network. Overall, the manuscript is clearly written, the proteomic and genomic data seems of high quality and appears to be properly analysed. As such, the manuscript represents a highly valuable resource that can provide novel insights into the function of Mediator. Therefore, and as the authors mention in the discussion, their data that can help to understand important topics in the field of transcriptional regulation, such as topological communication between enhancers and genes or recently proposed phase-separation models in transcriptional control. On the other hand, in its current format the paper is, as stated above, a valuable resource that provides mostly descriptive data, since the authors do not really attempt to experimentally test some of their proposed models. I think the paper would really benefit from some additional insights that aim at providing more mechanistic insights into the role of Mediator and/or its novel interacting partners in NSC:

- It is well established that Mediator is bound to most active enhancers in the different cell types where it has been investigated. However, what is still unclear is how Mediator is specifically recruited to such regulatory elements. The authors data suggest that master TFs could play a major role in this process but do not test this idea. The authors could perform different loss of

function and/or gain of function experiments to test their predictions: (i) generate KO lines for some of the candidate TFs and evaluate the effects on Mediator recruitment to enhancers; (ii) the previous strategy could lead to secondary effects (e.g. differentiation) that could obscure the interpretation of the results. The authors could delete instead the binding sites of their candidate TFs at selected enhancers and evaluate the consequences on Mediator recruitment; (iii) the authors could artificially tethered their TFs (e.g. dCas9 fusion) to particular genomic locations and see whether that is enough for Mediator recruitment.

- Can the authors disrupt Mediator in NSC and test whether this affects the recruitment of interacting TFs and chromatin remodellers to enhancers?. If not, is enhancer-gene communication affected?.

- The authors should confirm some of the Mediator-TF interactions using co-IP with and without Benzoase treatment. Do the TFs that interact with mediator share any common domains that could explain such interaction?. If so, the authors could delete such domain in at least one TF and evaluate whether the interaction with Mediator is lost.

- In my opinion the manuscript gives too much emphasis to superenhancers and broad promoters, which are two regulatory entities that are somehow based on artificial definitions. I think the authors should be more careful in the way they compare both of these regulatory elements and the genes associated to them: in how many of their Broad+SE genes do the SE and broad domains physically overlap?. The "improved" functional annotation of Broad+SE compared to SE-only genes could simply be due to a more accurate assignment to the right target gene for the former group. Therefore the authors should perhaps only considered SE that are truly distal from genes and that do not overlap transcription start sites. Moreover, the differences they observed between SE and classical enhancers could also be partly explained by SE being closer to their target genes and therefore being more accurately assigned to such targets. In this regard, what happens if the authors consider genes that are linked to not one but multiple classical enhancers (e.g. genes with 3 or more classical enhancers). Do such genes look more similar to SE genes?. At the end of the day, in many cases SE might "simply" represent a cluster of multiple classical enhancers that are close to each other.

Minor points:

- The authors could provide a more exhaustive list of previously known Mediator interactors and indicate which ones they observed and which ones they did not.

- I think that the overall description of the methods used to prepare their nuclear extracts and IPs is rather poor and should be considerably more detailed, as it could be useful for the scientific community.

- In fig 3B: is the number of TFs interacting with Mediator among SE genes statistically significant considering the 600 expressed TFs in NSC?.

- How are enhancers (classical or SE) assigned to their putative target genes?. Add to methods.

Reviewer #3 (Remarks to the Author):

Quevedo and colleagues employed an epitope-tagging purification method to identify proteins interacting with the Mediator complex in neural stem cells (NSCs). They identified key NSC transcription factors (TFs) and demonstrated genome-wide co-localization of Mediator and these TFs at enhancers with preferential binding at super-enhancer and broad H3K4me3-modified genes. The authors also investigated role of Tcf4 in activating transcription of other key NSC transcription factors. Overall, this is a well-done correlative study from both a technical and intellectual perspective. I believe the study would have a positive impact on the field.

However, while the Mediator interactome is interesting, this study does not reveal mechanistic information on the various TF-Mediator interactions, nor does it establish that the interactions are direct. It has been well documented that Mediator co-localizes with key TFs, and other factors at enhancers, particularly super-enhancers, in different cell lines. It is also known that key TFs are in a regulatory circuit to regulate cell identity; this topic has been explored extensively in ESCs, NSCs and NPCs. Even simple mechanistic analyses would strengthen this study considerably. For example, could the authors demonstrate by ChIP-seq that loss of Mediator (or associated factor)

occurs upon knockdown or degradation of one or two of the key TFs? Or could they perform biochemistry with a couple of pure TFs showing that they recruit pure Mediator to DNA? Anything to establish a direct and functional interaction.

Detailed comments:

Page 3, line 61. "Besides Mediator and Brd4, a number of chromatin modifiers, such as Ep300, Chd7, Smc1a (Cohesin complex), Brg1 (SWI-SNF complex), Chd4 (NuRD complex) and Kdm1a (LSD1 complex), were found to be enriched at SEs". Chd7 and Brg1 are chromatin remodelers rather than chromatin modifiers (Ho et al., 2010, Nature, Chromatin remodeling during development). Cohesin complex is neither a chromatin modifier nor remodeler in the traditional sense.

Page 4, line 74. "Mediator was shown to interact with SE-enriched chromatin modifiers Cohesin and Crebbp, suggesting ..." Again, Cohesin is not a chromatin modifier.

In Fig. 1c, a group of Med subunits were placed in the center of the diagram to represent Mediator complex and lines were used to show the interacting proteins. It's as if the figure is conveying the interactions between specific subunits and associated proteins. For example, it looks like Med12 interacts all the post-translational modification proteins, which is not the point of the manuscript. It would be better if the group of subunits is replaced by one big circle named Mediator. I would also like to know where the missing Mediator subunits are?

In Fig. 2a, it is not clear which nuclear extract (untreated, benzonase-treated, or EtBr-treated?) was used for IPs. Because the Mediator interactors were selected only when they co-purified with Mediator from both untreated and benzonase-treated NEs, it makes better sense if authors also show co-IPs performed using both NEs.

Page 7, line 146. "We identified 16 DNA sequence-specific transcription factors ..., including NFI TFs Nfia and Nfib, Sox2 and ..." The authors mentioned Sox2 is a novel Mediator interactor here. This is conflict with Fig. 1c, in which Sox2 is in grey, which indicates known interactors.

Page 7, line 148. "The majority of these TFs have an important function in the regulation of NSCs" What kind of regulation?

Page 11, line 235. "top 100 of each category" Does this mean 100 genes of the highest expression level in this category?

Fig. 5d-e. In Fig. 5e, it seems to me there is a super-enhancer spanning the whole Hes1 gene and its surrounding region. Thus in Fig. 5d, some signals may be from super-enhancers rather than promoters. It makes more sense to use genes whose super-enhancer is not overlapping its promoter for these analyses.

Fig. 6d. What about Tcf4 binding to SEs in other categories?

Reviewer #1.

1. One would wonder how similar is between the mouse neural stem cell line with the real neural stem cells in vivo. For cell line derived from iPSCs, the heterogeneity in cell population is also a problem. Since all the conclusions of this study came from the experiments using this cell line, these aforementioned questions should be addressed by comparing the transcriptome between this cell line with NSCs derived from mouse brain. Single cell sequencing or immunostaining experiments should be done to address the heterogeneity issue.

Our reply:

“Our” neural stem cell line was shown by the inventor of the differentiation protocol (Austin Smith) not to have an identical gene expression pattern to *in vivo* neural stem cells due to growth medium-induced changes (Pollard et al., 2008, Mol. Cell Neuroscience 38, 393-403). However, they are very well characterized to represent a homogenous adherent culture of cells that all stain positive for NSC markers Pax6 and RC2 (see for example our publications in Nature Genetics 2011 and Journal of Cell Biology 2009). Importantly, Austin Smith showed that these NSCs can still differentiate into the three major neural lineages; neurons, oligodendrocytes and astrocytes (Conti et al, 2005, PLoS Biology 3, e283, Glaser et al. 2007 PLoS One 2, e298) and Francois Guillemot showed that these NSCs go into reversible quiescence upon BMP4 addition (Martynoga et al., 2013, Genes Dev. 27, 1769-1786), similar to mouse NSCs *in vivo* (Mira et al. 2010, Cell Stem Cell 7, 78-89.). In conclusion, the NSCs that we use are not identical to *in vivo* NSCs but have preserved key features of *in vivo* NSCs and grow as homogenous adherent cultures.

2. For establishing the Flag-tagged MED15 NSCs, it should be checked to see how much of MED15 is incorporated into the endogenous Mediator complex using Co-IP or gel-filtration experiment. The exogenous expressed subunit of protein complex might be hard to integrate into the complex in vivo. Too much free-form of Flag-MED15 could skew the Co-IP plus mass spec results.

Our reply:

We addressed this point of the reviewer by performing large-scale endogenous Mediator IPs by Med12 antibody and control IPs, with or without benzonase, followed by mass spectrometry to analyze the composition of endogenous Mediator and its interactors and compare it to the FLAG-Med15 interactome. EmPAI scores provide a good indication of relative protein quantity. As can be seen by comparing Supplementary Table 1 with the FLAG-Med15 data and the new Supplementary Table 2 with the Med12 IP data, FLAG-Med15 has an about 2-fold higher emPAI score than Med15 in the endogenous Mediator, whereas the remainder of the Mediator subunits is about 2-fold lower in the FLAG-Med15 IP as compared to the purified endogenous Mediator. We conclude that about a quarter of FLAG-Med15 is incorporated into the Mediator complex. Importantly, we find 60 of the 96 FLAG-Med15 Mediator interactors to be again specifically present in the endogenous Med12 IP. Given the lower sensitivity and higher background of antibody IPs (indeed 22 FLAG-Med15 interactors were “lost” in the Med12 IP due to high background in the controls, see new Supplementary Table 2), we consider this a very high number that further validates our FLAG-Med15 IP data.

We say in our revised manuscript (page 7-8): To validate our FLAG-affinity approach, we also purified endogenous Mediator from NSCs by immunoprecipitation with a Med12 antibody (Supplementary Table

2). We find back 60 of the 96 interactors identified in FLAG-Mediator purifications, including 11 transcription factors. With the lower sensitivity and higher background generally observed in endogenous IPs, we consider this number of overlapping Mediator interactors a validation of our FLAG-Mediator purifications.

3. It is interesting that the Mediator associated partners in NSCs include RNA binding factors. Mediator was reported to be involved in mRNA processing, which might be relevant and should be discussed.

We thank the reviewer to attend us to this point. We now say in our revised manuscript (page 7):

One prominent Mediator interactor category is mRNA binding proteins (Fig. 1c). We find that Mediator interacts with alternative splicing regulators Hnrnpf and Mbnl1 and cleavage and polyadenylation factors Cpsf1 and Cpsf2. These interactions may facilitate the role that Mediator plays in regulating alternative splicing and alternative cleavage and polyadenylation of pre-mRNAs²⁸. (Ref28: Huang, Y. et al. Mediator complex regulates alternative mRNA processing via the MED23 subunit. Mol Cell 45, 459-69 (2012)).

4. Through this investigation, a Mediator interaction network of enhancer- and promoter- binding proteins, however, all these protein partners are not necessarily in direct contact with the Mediator complex. Association rather than interaction could be most of the cases, however, the authors have used “interaction” throughout the whole manuscript.

Our reply:

The reviewer is correct that we cannot prove direct contact between Mediator and its identified interactors. Indeed, we never claim such direct contacts. Nevertheless, interaction (direct or indirect) is the commonly used term for the type of interactions that we here identify, for example in the often-used term “protein-protein-interaction (PPI) networks”. We feel that “interaction” is the least inaccurate term in this respect. The suggested “association” is a more wide term that is often used for (non-physical) correlations (for example in Genome-wide association studies), which defines very different data from the ones we generate here. We have now inserted the sentence (page 6) “Mediator interacting proteins may interact with Mediator directly or via other proteins” to indicate the reviewer’s point that we cannot be sure of direct contact between Mediator and its interactors.

Reviewer #2

What is still unclear is how Mediator is specifically recruited to such regulatory elements. The authors data suggest that master TFs could play a major role in this process but do not test this idea. The authors could perform different loss of function and/or gain of function experiments to test their predictions:

Our reply:

We have now performed large-scale RNAi-mediated knock-down (144 transfections of NSCs by electroporation in total in the biological duplicate experiments) of master TFs Tcf4 or Sox2 in NSCs (knock-down values are in new Figure 5a) and performed Mediator ChIP on several enhancers where genome binding of Tcf4 and Sox2 coincides with Mediator genome binding (selected from the ChIP-seq data of the respective factors). We first show by Med12 ChIP-RT-PCR that Mediator is indeed highly

enriched at the selected sites (new Figure 5b). We then show that knock-down of Tcf4 significantly reduces Mediator recruitment at all five selected sites (new Figure 5c). Knock-down of Sox2 significantly reduces Mediator recruitment at enhancers 6.7 kb upstream from the *Olig1* gene and 6 kb in the *Tulp3* gene (new Figure 5d). We find that Mediator binding at 30 kb downstream of the *Olig1* gene, 8.6 kb in the *Klf15* gene and 6.5 kb in the *Jag1* gene is not significantly affected by Sox2 knock-down (new Figure 5d).

We say in the revised text (page 10-11):

We tested whether genome recruitment of Mediator depends on some of its interacting TFs. We performed shRNA-mediated knock-down for TFs Tcf4 or Sox2 (Fig. 5a). We selected a number of enhancers from our ChIP-seq data for Mediator, Tcf4 and Sox2 where Mediator genome binding overlaps with genome binding by Tcf4 and Sox2. We find by Med12 ChIP RT-PCR that Mediator is indeed highly enriched at the selected sites (Fig. 5b). Knock-down of Tcf4 significantly reduced Mediator binding at all five selected sites (Fig. 5c). Knock-down of Sox2 significantly reduced Mediator binding at enhancers 6.7 kb upstream from *Olig1* and 6 kb in *Tulp3* (Fig. 5d). We find that Mediator binding at 30 kb downstream of *Olig1*, 8.6 kb in *Klf15* and 6.5 kb in *Jag1* are not significantly affected by Sox2 knock-down (Fig. 5d). We conclude that efficient Mediator recruitment to individual genomic sites can depend on its interaction partners Tcf4 or Sox2.

In the discussion (page 16-17) we have inserted the sentence: Accordingly, we find that Tcf4 and Sox2 are required for optimal Mediator recruitment to some of the tested genomic sites where the three factors have overlapping binding.

In the Methods we describe the new experiments (page 24).

Can the authors disrupt Mediator in NSC and test whether this affects the recruitment of interacting TFs and chromatin remodellers to enhancers?. If not, is enhancer-gene communication affected?.

Our reply:

This is a very difficult experiment. Knocking-down individual Mediator subunits may either cause cell death, if they disrupt Mediator complex structure as Mediator is essential for gene regulation, or may not affect recruitment of other factors as the targeted Mediator subunit may not provide the required interaction surface. Currently we do not know which Mediator subunits provide the interaction surface(s) for the different interaction partners that we identified.

The authors should confirm some of the Mediator-TF interactions using co-IP with and without Benzonase treatment. Do the TFs that interact with mediator share any common domains that could explain such interaction?. If so, the authors could delete such domain in at least one TF and evaluate whether the interaction with Mediator is lost.

Our reply:

We have performed the requested experiment. We performed IP of endogenous Med12 with or without benzonase from neural stem cell nuclear extracts, plus an IgG control IP with or without benzonase, and compared it to the FLAG-Med15 interactome (new Supplementary Table 2). We find back 13 of the original 16 FLAG-Med15 interacting TFs in the endogenous Med12 IP, of which 2 are now also in the background. Of the remaining 11 TFs, 8 are again not affected by benzonase (by the 2-fold difference in emPAI score cut-off, that we use in the paper), including Tcf4 and Sox2. The remaining 3 TFs are either

near the 2-fold threshold (Tcf12, Znf24) or of low amount in the first place (Trps1). We did not find any common domains between the Mediator-interacting TFs. We say in the revised text (page 8):

To validate our FLAG-affinity approach, we also purified endogenous Mediator from NSCs by immunoprecipitation with a Med12 antibody (Supplementary Table 2). We find back 60 of the 96 interactors identified in FLAG-Mediator purifications, including 11 transcription factors. With the lower sensitivity and higher background generally observed in endogenous IPs, we consider this number of overlapping Mediator interactors a validation of our FLAG-Mediator purifications.

The "improved" functional annotation of Broad+SE compared to SE-only genes could simply be due to a more accurate assignment to the right target gene for the former group. Therefore the authors should perhaps only considered SE that are truly distal from genes and that do not overlap transcription start sites.

Our reply:

SEs, as defined by the previously published algorithm ROSE, do not cross with a window of 1 kb on either side of the TSS. This was not explained in the methods, which we have now corrected (page 25). Therefore, all SEs are distal from the TSS, by definition.

Moreover, the differences they observed between SE and classical enhancers could also be partly explained by SE being closer to their target genes and therefore being more accurately assigned to such targets. In this regard, what happens if the authors consider genes that are linked to not one but multiple classical enhancers (e.g. genes with 3 or more classical enhancers). Do such genes look more similar to SE genes?. At the end of the day, in many cases SE might "simply" represent a cluster of multiple classical enhancers that are close to each other.

Our reply: We have tested whether SEs are closer to the TSS of the nearest gene than classical (typical) enhancers. We find (Figure 1 for the reviewer, below) that typical enhancers have a similar distance distribution from the TSS (either from a non-broad or broad promoter) than SEs have. If multiple typical enhancers (3 or more typical enhancers that are nearest to the tested gene TSS and not to another TSS) map to a TSS, the nearest of these enhancers have a similar distance distribution to the TSS as SEs have. We then tested whether broad or non-broad promoters have a different distribution of RNAPol2, Mediator (Med1), Integrator (Ints11), Tcf4 + Sox2 + NFI or Chd7, Jmjd1c or Carm1 depending on whether their TSS is nearest to an SE, one typical enhancer or multiple classical (typical) enhancers. We find (Figure 2 for the reviewer, below) that all (broad and non-broad) promoters that are nearest to an SE have a broader signal for all the above factors, as compared to promoters that are nearest to one or multiple typical enhancers. For RNA pol2, Mediator and Integrator the signal is also higher at the TSS of promoters nearest to an SE. Non-broad promoters have near-identical coverage of all the factors irrespective of whether they are nearest to one or multiple typical enhancers, but a lower and less broad coverage for all factors as compared to non-broad promoters near an SE. We also tested the expression of genes associated with the different categories of promoters and enhancers (Figure 3 for the reviewer, below). We find no difference in expression between broad promoters nearest to a typical enhancer (Broad-SE) and broad promoters near multiple typical enhancers (Multiple+Broad), as well as between non-broad promoters of these two categories (Typical vs. Multiple-Broad). Broad promoters nearest to an SE (SE+Broad) still have a higher expression than all the other categories.

These plots suggest that SEs do infer special features at promoters and genes near them that are not recapitulated by multiple classical/typical enhancers.

% of enhancers in each distance range

Figure 1 Distance from TSS of enhancers in different categories. Typical_multiple = 3 or more typical enhancers closest to this TSS.

Figure 2 Binding (ChIP-seq signal) distribution of the indicated factors/ histone modifications at the TSS area of different enhancer-promoter combinations. Distance from the TSS (in bp) is on the X-axis. Multiple = 3 or more typical enhancers closest to this TSS.

Figure 3 Expression levels of genes with enhancers and promoters in the different categories. Multiple = 3 or more typical enhancers closest to this TSS.

Minor points

The authors could provide a more exhaustive list of previously known Mediator interactors and indicate which ones they observed and which ones they did not.

We searched in the BIOGRID protein interaction database for interactors of human Mediator subunits that are not Mediator itself. We found 31 Mediator subunits with together several hundred interactors. However, outside known Mediator interactors such as the RNAPol2 complex (that we also report in this manuscript), we find very little overlap in the putative interactors between different Mediator subunits. As these subunits should co-purify as one highly stable Mediator complex, one would expect to see significant overlap between their respective interactors, if these were true interactors. Moreover, reported Mediator interactors are frequently cytoplasmic proteins, which is another reason to have concerns on the reliability of these interaction data. Together, we doubt whether many of these are true Mediator interactors and are hesitant to include them in our paper as it would give them a respectability as a Mediator interactor that we feel many of them do not deserve.

We have reported in Figure 1c all the interactors that we observe and (in grey) which ones were previously identified.

I think that the overall description of the methods used to prepare their nuclear extracts and IPs is rather poor and should be considerably more detailed, as it could be useful for the scientific community.

The used methodology is described in great detail for nuclear extract preparation in Dignam et al (1983) and the FLAG-affinity purification and the antibody-mediated IP in our previous papers in Cell Stem Cell (Van den Berg et al. 2010) and Nature Genetics (Engelen et al. 2011). We cite these papers in this context in the methods of the current manuscript. Van den Berg et al. is particularly detailed as the paper is about presenting the methods. Engelen et al. subsequently applies the technology in NSCs. We follow these protocols to the letter. A more elaborate description would just be an exact copy of the methods in these papers. We have now described the Med12-IP in the methods in greater detail, as it slightly deviates in methodology from our earlier protocols.

Reviewer #3

For example, could the authors demonstrate by ChIP-seq that loss of Mediator (or associated factor) occurs upon knockdown or degradation of one or two of the key TFs? Or could they perform biochemistry with a couple of pure TFs showing that they recruit pure Mediator to DNA? Anything to establish a direct and functional interaction.

Our reply:

We have now performed large-scale RNAi-mediated knock-down (144 transfections of NSCs by electroporation in total in the biological duplicate experiments) of key TFs Tcf4 or Sox2 in NSCs (knock-down values are in new Figure 5a) and performed Mediator ChIP on several enhancers where genome binding of Tcf4 and Sox2 coincides with Mediator genome binding (selected from the ChIP-seq data of the respective factors). We first show by Med12 ChIP-RT-PCR that Mediator is indeed highly enriched at the selected sites (new Figure 5b). We then show that knock-down of Tcf4 significantly reduces Mediator recruitment at all five selected sites (new Figure 5c). Knock-down of Sox2 significantly reduces Mediator recruitment at enhancers 6.7 kb upstream from the *Olig1* gene and 6 kb in the *Tulp3* gene (new Figure

5d). We find that Mediator binding at 30 kb downstream of the *Olig1* gene, 8.6 kb in the *Klf15* gene and 6.5 kb in the *Jag1* gene is not significantly affected by Sox2 knock-down (new Figure 5d).

We say in the revised text (page 10-11):

We tested whether genome recruitment of Mediator depends on some of its interacting TFs. We performed shRNA-mediated knock-down for TFs Tcf4 or Sox2 (Fig. 5a). We selected a number of enhancers from our ChIP-seq data for Mediator, Tcf4 and Sox2 where Mediator genome binding overlaps with genome binding by Tcf4 and Sox2. We find by Med12 ChIP RT-PCR that Mediator is indeed highly enriched at the selected sites (Fig. 5b). Knock-down of Tcf4 significantly reduced Mediator binding at all five selected sites (Fig. 5c). Knock-down of Sox2 significantly reduced Mediator binding at enhancers 6.7 kb upstream from *Olig1* and 6 kb in *Tulp3* (Fig. 5d). We find that Mediator binding at 30 kb downstream of *Olig1*, 8.6 kb in *Klf15* and 6.5 kb in *Jag1* are not significantly affected by Sox2 knock-down (Fig. 5d). We conclude that efficient Mediator recruitment to individual genomic sites can depend on its interaction partners Tcf4 or Sox2.

In the discussion (page 16-17) we have inserted the sentence: Accordingly, we find that Tcf4 and Sox2 are required for optimal Mediator recruitment to some of the tested genomic sites where the three factors have overlapping binding.

In the Methods we describe the new experiments (page 24).

Page 3, line 61. “Besides Mediator and Brd4, a number of chromatin modifiers, such as Ep300, Chd7, Smc1a (Cohesin complex), Brg1 (SWI-SNF complex), Chd4 (NuRD complex) and Kdm1a (LSD1 complex), were found to be enriched at SEs”. Chd7 and Brg1 are chromatin remodelers rather than chromatin modifiers (Ho et al., 2010, Nature, Chromatin remodeling during development). Cohesin complex is neither a chromatin modifier nor remodeler in the traditional sense.

Our reply:

We have changed the sentence into:

Besides Mediator and Brd4, chromatin modifiers such as Ep300 and Kdm1a (LSD1 complex), chromatin remodelers such as Chd7, Brg1 (SWI-SNF complex) and Chd4 (NuRD complex) and Smc1a (Cohesin complex) were found to be enriched at SEs.

Page 4, line 74. “Mediator was shown to interact with SE-enriched chromatin modifiers Cohesin and Crebbp, suggesting ...” Again, Cohesin is not a chromatin modifier.

Changed into: *Mediator was shown to interact with SE-enriched chromatin modifier Crebbp¹⁸ and the Cohesin complex¹⁹, suggesting...*

In Fig. 1c, a group of Med subunits were placed in the center of the diagram to represent Mediator complex and lines were used to show the interacting proteins. It’s as if the figure is conveying the interactions between specific subunits and associated proteins. For example, it looks like Med12 interacts all the post-translational modification proteins, which is not the point of the manuscript. It would be better if the group of subunits is replaced by one big circle named Mediator. I would also like to know where the missing Mediator subunits are?

Our reply:

The reviewer is correct that our current Fig. 1c could be misleading. As suggested, we have encircled the Mediator complex and let interaction run to the edge of the circle to indicate that we do not know which Mediator subunit is interacting.

We find 27 subunits of Mediator. Med1, 4, 6, 8, 10, 12, 13, 13l, 14, 15, 16, 17, 18, 19, 20, 22, 23, 24, 25, 26, 27, 28, 29, 30, 31, Ccnc and Cdk19 (the neural lineage replacement of Cdk8)

Med2 and Med3 do not exist in mammalian cells (Bourbon et al. (2004) A unified nomenclature of protein subunits of Mediator complexes linking transcriptional regulators to RNA polymerase II. *Mol. Cell* 14, 553-557). Med7, 9, 11 and 21 were very low and not present in all purified FLAG-Mediator samples. Med12L was never observed in our Mediator purifications and is indeed very lowly expressed in our NSCs (our RNA-seq data).

In Fig. 2a, it is not clear which nuclear extract (untreated, benzonase-treated, or EtBr-treated?) was used for IPs. Because the Mediator interactors were selected only when they co-purified with Mediator from both untreated and benonase-treated NEs, it makes better sense if authors also show co-IPs performed using both NEs.

In the Methods we refer to our 2010 paper in *Cell Stem Cell* in which the IPs are treated with benzonase. Accordingly, our Carm1 and Jmjd1c IPs were treated with benzonase. We have now extended our Methods on these IPs to include this and other details.

We have now performed a large-scale Med12 antibody IP of endogenous Mediator with and without benzonase treatment and analyzed the interactors by mass spectrometry, which allows us to judge all our previously identified FLAG-Mediator interactors (new Supplementary Table 2). We find 60 of the 96 FLAG-Med15 Mediator interactors to be again specifically present in the endogenous Med12 IP. Given the lower sensitivity, higher variability and higher background of antibody IPs (indeed 22 FLAG-Med15 interactors were “lost” in the Med12 IP due to high background in the controls) this is a very high number that further validates our FLAG-Med15 IP data. We find that only 6 out the 60 specific interactors were not detected in the benzonase-treated Med12 IP, which considering the higher variability of antibody-IPs, we believe provides a good validation of our initial FLAG-Mediator interactor selection.

We say in the revised text (page 8):

To validate our FLAG-affinity approach, we also purified endogenous Mediator from NSCs by immunoprecipitation with a Med12 antibody (Supplementary Table 2). We find back 60 of the 96 interactors identified in FLAG-Mediator purifications, including 11 transcription factors. With the lower sensitivity and higher background generally observed in endogenous IPs, we consider this number of overlapping Mediator interactors a validation of our FLAG-Mediator purifications.

Page 7, line 146. “We identified 16 DNA sequence-specific transcription factors ..., including NFI TFs Nfia and Nfib, Sox2 and ...” The authors mentioned Sox2 is a novel Mediator interactor here. This is conflict with Fig. 1c, in which Sox2 is in grey, which indicates known interactors.

The reviewer is correct that this sentence is ambiguous. We have changed this to:

We identified 16 DNA sequence-specific transcription factors (TFs) of which 14 are novel Mediator interactors (Fig. 1c). Identified TFs include NFI TFs Nfia and Nfib, Sox2 and E-box TFs Tcf4 and Tcf12.

Page 7, line 148. “The majority of these TFs have an important function in the regulation of NSCs” What kind of regulation?

The type of regulation of NSCs is indicated in Figure 2c, mostly NSC self-renewal or NSC differentiation.

Page 11, line 235. “top 100 of each category” Does this mean 100 genes of the highest expression level in this category?

We did not explain what we mean by top 100, for which we apologize. We have now included in the Methods the following explanation (page 27): Top 100 SE+Broad genes are genes with a broad promoter and from that category the 100 genes that are the nearest gene to the highest SEs (i.e. one gene per SE), as ranked by Mediator signal. Top 100 SE-Broad genes are genes without a broad promoter and from that category the 100 genes that are the nearest gene to the highest SEs, as ranked by Mediator signal. Top 100 Broad-SE genes are genes that not the nearest gene to an SE and from that category the 100 genes ranked by the broadest H3K4me3 signal at their promoter, from an already described list of mouse NSC broad H3K4me3 promoters³⁶ (Ref36: Benayoun, B.A. et al. H3K4me3 breadth is linked to cell identity and transcriptional consistency. *Cell* **158**, 673-88 (2014).)

Fig. 5d-e. In Fig. 5e, it seems to me there is a super-enhancer spanning the whole *Hes1* gene and its surrounding region. Thus in Fig. 5d, some signals may be from super-enhancers rather than promoters. It makes more sense to use genes whose super-enhancer is not overlapping its promoter for these analyses.

The Med1 bound region across the *Hes1* gene does not constitute an SE. The *Hes1* gene has an SE but it is 100 kb downstream. SEs, as defined by the previously published algorithm ROSE, do not cross with a window of 1 kb on either side of a TSS. This was not explained in the methods, which we have now corrected (page 25). Therefore, all SEs are distal from the TSS, by definition. Nevertheless, some of the Med1 peaks outside the *Hes1* gene may be at enhancers, which confuses the point we are trying to make about the overlap between Med1 and its interactors at individual broad promoters. We have therefore chosen another broad promoter, of the *Trim8* gene, where ChIP-seq signal for H3K4me3, Med1, RNAPol2 and the Mediator interactors highly overlap (new Figure 6e).

Fig. 6d. What about *Tcf4* binding to SEs in other categories?

We have now included in a new Figure 7d *Tcf4* binding to SE-Broad genes (the other SE category) and find it to be 77%. We now say (page 13): Indeed, *Tcf4* is present on nearly all SEs of SE+Broad and SE-Broad genes (Fig. 7d).

Reviewers' comments:

Reviewer #1 (Remarks to the Author):

This reviewer has carefully gone through the revised manuscript, and felt that the authors have addressed all the raised questions satisfactorily. Congratulations!!!

Reviewer #2 (Remarks to the Author):

I thank the authors for their extensive revision and clear explanations in the rebuttal letter. The authors have successfully addressed most of my initial concerns. However, in my opinion, one of my major points has not been properly addressed:

- Can the authors disrupt Mediator in NSC and test whether this affects the recruitment of interacting TFs and chromatin remodellers to enhancers?. If not, is enhancer-gene communication affected?.

Our reply:

This is a very difficult experiment. Knocking-down individual Mediator subunits may either cause cell death, if they disrupt Mediator complex structure as Mediator is essential for gene regulation, or may not affect recruitment of other factors as the targeted Mediator subunit may not provide the required interaction surface. Currently we do not know which Mediator subunits provide the interaction surface(s) for the different interaction partners that we identified.

As for Mediator, the constitutive loss of CTCF or the Cohesin complex results in cell death. In a couple of recent papers, the role of these proteins on 3D genome organization was investigated by using the AID degron system in order to rapidly and inducibly degrade these proteins (for the Cohesin complex, the RAD21 subunit was targeted) and thus overcome the cell death problem (Nora et al, Cell; Rao et al, Cell). As the NSC used by the authors are derived from ESC, which are genetically highly tractable, the authors could in principle use the AID system to target a Mediator subunit that disrupts the whole complex. This experimental system should allow the authors to address my previous questions. Alternatively, the authors could use an inducible shRNA approach to knockdown a key Mediator subunit. In my opinion, experimentally testing whether Mediator controls the recruitment of other regulatory factors to enhancers and/or enhancer-gene 3D communication could really improve the manuscript.

Reviewer #3 (Remarks to the Author):

Understanding Mediator interactions genomewide is an important problem in the field. This paper does a nice comparison of Mediator-proteomics in NSCs to genomewide co-localization of mediator with chromatin enzymes and NSC TFs. In the rebuttal, the authors made the appropriate changes in response to my comments and experimentally or textually addressed most of my questions. The manuscript is improved. There are a few minor issues that could have been addressed to further strengthen.

1. The efficiency of NSC generation from pluripotent stem cells varies depending on their cell state and culture system even though it is performed according to a good published protocol. The heterogeneity in the in cell population is unknown. Showing immunostaining results of Sox2, Nestin or other markers of the neural stem cells, with quantification of the percentage of the Sox2 and/or Nestin in the cell population, would have strengthened the claims albeit it is not a major driver of my review.

2. The authors showed reduced Mediator binding at the regulatory elements of selected genes upon Tcf4 and Sox2 knockdown. Because the Sox2 and Tcf4 are significantly enriched in the Med12 purification, either Sox2-Mediator or Tcf4-Mediator interaction may be involved in a large number of the gene regulation events. The results of the selected gene testing only partially supports the conclusion. If the selected genes are significantly regulated by TF-Mediator

interactions, they should perform genomewide CHIP of Med1 in the knockdowns and correlate with gene expression by proximity rule.

I found no major problem with the stats, scripts, or rigor. The analyses are described in enough detail here and in previous papers so as to allow adequately attempts to reproduce.

There's a lot of good data here and I'm happy to support the paper.

Reviewer #2

As for Mediator, the constitutive loss of CTCF or the Cohesin complex results in cell death. In a couple of recent papers, the role of these proteins on 3D genome organization was investigated by using the AID degenon system in order to rapidly and inducibly degrade these proteins (for the Cohesin complex, the RAD21 subunit was targeted) and thus overcome the cell death problem (Nora et al, Cell; Rao et al, Cell). As the NSC used by the authors are derived from ESC, which are genetically highly tractable, the authors could in principle use the AID system to target a Mediator subunit that disrupts the whole complex. This experimental system should allow the authors to address my previous questions. Alternatively, the authors could use an inducible shRNA approach to knockdown a key Mediator subunit. In my opinion, experimentally testing whether Mediator controls the recruitment of other regulatory factors to enhancers and/or enhancer-gene 3D communication could really improve the manuscript.

Our reply: In our paper, we focus on novel Mediator interactors and their potential role in recruiting Mediator to the genome. The proposed experiments are not only technically demanding and would require many months of work, they also do not address the recruitment of Mediator. The impact of Mediator-interactor interactions on promoter-enhancer 3D communication is outside our expertise and the scope of this manuscript.

Reviewer #3

There are a few minor issues that could have been addressed to further strengthen.

1. The efficiency of NSC generation from pluripotent stem cells varies depending on their cell state and culture system even though it is performed according to a good published protocol. The heterogeneity in the in cell population is unknown. Showing immunostaining results of Sox2, Nestin or other markers of the neural stem cells, with quantification of the percentage of the Sox2 and/or Nestin in the cell population, would have strengthened the claims albeit it is not a major driver of my review.

Our reply: We have now performed antibody-mediated stainings of our NSCs for NSC markers Sox2 and Nestin. As shown in new Supplementary Figure 3a and 3b, we find that essentially 100% of our NSCs stain positive for both Sox2 and Nestin. This confirms that we work with a homogenous population of NSCs.

We say (page 20): Essentially all our NSCs express NSC markers Sox2 and Nestin (Supplementary Fig. 3a and 3b).

2. The authors showed reduced Mediator binding at the regulatory elements of selected genes upon Tcf4 and Sox2 knockdown. Because the Sox2 and Tcf4 are significantly enriched in the Med12 purification, either Sox2-Mediator or Tcf4-Mediator interaction may be involved in a large number of the gene regulation events. The results of the selected gene testing only partially supports the conclusion. If the selected genes are significantly regulated by TF-Mediator interactions, they should perform genomewide ChIP of Med1 in the knockdowns and correlate with gene expression by proximity rule.

Our reply: This experiment is logistically not possible. For one such quantitative Mediator ChIP-seq experiment (which needs to be performed in duplicate to be reliable) one needs 600 micrograms of chromatin (TF knock-down Mediator ChIP and a control knock-down Mediator ChIP). This is an order of magnitude higher amount than what we used for the current small-scale knock-down ChIPs in Figure 5 in

the paper and would require 720 individual electroporations of pSuper plasmid DNAs into NSCs in 720 individual dishes followed by selection and collection. And with neural stem cells, which are less robust than commonly used cell lines, manipulation time matters, making the proper handling of so many samples within one experiment not feasible.

Although this experiment could generate a more comprehensive picture, it will not change our finding (Figure 5) that knock-down of Tcf4 or Sox2 can affect Mediator recruitment to individual Sox2 and Tcf4 genome targets.